# Neural basis of corruption in power-holders

**Yang Hu[1,2†], Chen Hu[3,4], Edmund Derrington[2,5], Brice Corgnet[6], Chen Qu[1]*, Jean-Claude Dreher[2,5]**

[1]Key Laboratory of Brain, Cognition and Education Sciences, Ministry of Education, China; School of Psychology, Center for Studies of Psychological Application, and Guangdong Key Laboratory of Mental Health and Cognitive Science, South China Normal University, Guangzhou, China; [2]Laboratory of Neuroeconomics, Institut des Sciences Cognitives Marc Jeannerod, CNRS, Lyon, France; [3]Motivation, Brain & Behavior (MBB) Team, Institut du Cerveau et Moelle Epiniere, Hôpital de la Pitié-Salpêtrière, Paris, France; [4]Sorbonne Université, Paris, France; [5]Université Claude Bernard Lyon 1, Lyon, France; [6]EmLyon, Ecully, France

**Abstract** Corruption often involves bribery, when a briber suborns a power-holder to gain advantages usually at a cost of moral transgression. Despite its wide presence in human societies, the neurocomputational basis of bribery remains elusive. Here, using model-based fMRI, we investigated the neural substrates of how a power-holder decides to accept or reject a bribe. Power-holders considered two types of moral cost brought by taking bribes: the cost of conniving with a fraudulent briber, encoded in the anterior insula, and the harm brought to a third party, represented in the right temporoparietal junction. These moral costs were integrated into a value signal in the ventromedial prefrontal cortex. The dorsolateral prefrontal cortex was selectively engaged to guide anti-corrupt behaviors when a third party would be harmed. Multivariate and connectivity analyses further explored how these neural processes depend on individual differences. These findings advance our understanding of the neurocomputational mechanisms underlying corrupt behaviors.

*For correspondence: fondest@163.com

Present address: †School of Psychological and Cognitive Sciences, Peking University, Beijing, China

Competing interests: The authors declare that no competing interests exist.

## Introduction

Corruption, commonly defined as 'the abuse of entrusted power for private gains' (*Graycar and Smith, 2011*), is one of the most pervasive and complex social problems today. It distorts development priorities, aggravates inequality in societies, and challenges organizations and governments globally (*Mauro, 1995*). Due to its critical societal implications, corruption has been extensively investigated during the past two decades in fields such as political science, sociology, economics, and psychology. Using survey-based measures (*Lambsdorff, 2006*) and behavioral economic experiments (*Serra and Wantchekon, 2012*), these studies significantly improved our understanding of corruption at sociological and behavioral levels. However, the neural mechanisms underlying decision-making involving corrupt acts remain unexplored.

One of the most common forms of corruption is bribery, which costs between 1500 and 2000 billion US dollars each year (~2% of the global GDP) (*Gaspar and Hagan, 2016*). It usually occurs in an interpersonal context in which at least two parties are involved, a briber and a power-holder. The briber often pays the power-holder in some way, and, in return, the power-holder makes decisions that profit the briber (*Abbink, 2006*; *Köbis et al., 2016*). Critically, favor exchanges can hardly be considered as bribes unless they involve moral or legal transgression. That is, a bribe takes place when the briber obtains a favorable treatment or an advantageous outcome by paying the power-holder to circumvent a norm of impartiality or a moral rule (*Köbis et al., 2016*; *Serra and*

*Wantchekon, 2012*). In addition, bribery often implies harming the interests of a third party, which can be an explicit person or a societal organization (*Barr and Serra, 2009*; *Köbis et al., 2016*). For instance, a public official who accepts a bribe regarding a procurement contract will harm competitors whose project might bring greater benefits to society. Thus, one can distinguish two forms of moral costs incurred by the power-holder who takes a bribe: conniving with a fraud committed by the briber and harming an innocent third party. Indeed, previous behavioral studies have identified a crucial role of moral cost of unethical behaviors in explaining why individuals often refrain from behaving immorally. For example, people are reluctant to tell a self-serving lie or inflict physical harm on an anonymous partner for their own benefit even when there is no material cost of doing so (*Crockett et al., 2017*; *Rosenbaum et al., 2014*).

To understand the neural bases of the power-holder's corrupt behavior, it is necessary to start by asking how different moral costs are represented at the neural level while deciding whether to take a bribe or not. Previous fMRI studies in the moral domain provide valuable insights to formulate hypotheses regarding the neural underpinnings of moral costs in the context of bribery. On the one hand, the ventral part of anterior insula (vAI) has been shown to be engaged when moral and social norms are violated. For instance, a stronger vAI signal has been observed when people are deceived by another person (*Yin and Weber, 2016*) or when deciding whether to lie to obtain additional gains (*Yin et al., 2017*). A natural hypothesis is that the vAI is more engaged in representing the expected 'dirty' personal gains from accepting bribes, which may elicit a negative affective signal due to the moral cost of colluding with the briber who is violating the moral principle of honesty. On the other hand, the temporoparietal junction (TPJ) has been found to contribute to balance personal interests and other's welfare (*Hutcherson et al., 2015*; *Morishima et al., 2012*; *Obeso et al., 2018*). This region is also more active when one's decision impacts a person who is in a disadvantageous situation. For instance, a recent fMRI study found that multivoxel patterns of TPJ can discriminate generous from selfish choices during charitable decision-making (i.e., donating to help someone else in desperate need) (*Tusche et al., 2016*). Another study showed that the posterior part of TPJ encoded the levels of other's need when individuals decided whether to help another person (*Hu et al., 2017*). These findings suggested that the TPJ would be sensitive to the moral cost of the bribe-induced financial losses incurred by a third party.

A key question is to understand how these moral costs are integrated with other decision components into a neural value signal during bribery-related decision-making. Here, we developed and tested a number of computational models that assume that a power-holder makes the decision to accept (or reject) a bribe by computing the subjective value (SV) of each option *via* a utility function. Under the framework of social preference theory (*Fehr and Krajbich, 2014*), this function takes into account not only the trade-off between one's own gain and the briber's gain but also any moral costs specifically brought by taking the bribe. Using a model-based fMRI approach (*O'Doherty et al., 2007*), we searched for brain regions that were involved in computing such SV signals when deciding whether to accept a bribe. Since the ventromedial prefrontal cortex (vmPFC) is known to be the hub of value computation for both non-social (*Bartra et al., 2013*) and social decisions (*Ruff and Fehr, 2014*), we predicted the recruitment of vmPFC in computing an integrative decision value signal preceding the decision of whether to accept a bribe.

We also aimed to identify which brain region(s) guide the specific choices of taking a bribe or not. A recent theory argues that the dorsolateral prefrontal cortex (dlPFC) is key to support the pursuit of moral goals in a context-dependent manner (*Carlson and Crockett, 2018*). Hence, the dlPFC can be expected to flexibly align with specific bribery scenarios that concern different types of moral costs and regulate specific choices. In particular, the dlPFC plays a critical role in causally modulating self-serving dishonest behaviors (*Maréchal et al., 2017*; *Zhu et al., 2014*) and norm-enforcing decisions (*Buckholtz et al., 2015*; *Knoch et al., 2006*). A recent fMRI study also identified selective recruitment of the dlPFC in evaluating ill-gotten money obtained by harming others (vs. self) (*Crockett et al., 2017*). More intriguingly, recent work has highlighted the modulatory role of dlPFC on the value signal in vmPFC especially when the decision-making process requires individuals to exert self-control to inhibit the impulse to choose immediate rewards (vs. long-term rewards; *Hare et al., 2009*; *Hare et al., 2014*) or personal profits (vs. moral values; *Baumgartner et al., 2011*; *Dogan et al., 2016*). Here, we tested whether the functional connectivity between vmPFC and dlPFC would be enhanced during bribery-related decision-making. Based on the evidence above, we hypothesized that we would observe a stronger vmPFC–dlPFC functional coupling during

the decisions of whether to accept or reject a bribe (vs. offers in the Control condition) in which participants as power-holders might need more self-control to overcome the lure of accepting bribes that result in moral costs.

Our last question was to examine how individual differences in neural activities during bribery-related decision-making explain variations across power-holders, in preferences for bribery, characterized by moral costs due to bribe-taking. A growing body of literature links the dlPFC signal to individual variations in the levels of self-serving dishonesty (*Dogan et al., 2016*; *Yin and Weber, 2018*) or harm aversion (*Crockett et al., 2017*), which are closely related with two types of moral costs specific to corrupt acts measured in the current task. Hence, we performed an exploratory analysis, again with a focus on the dlPFC given the evidence above, to probe whether such a relationship exists, in a bribery setting, by applying a multivariate approach. We performed an inter-subject representational similarity analysis (IS-RSA) to uncover the neural–behavioral relationship across individuals (*van Baar et al., 2019*). Compared with the mass-univariate approach, IS-RSA allows us to associate multidimensional behavioral measures with a geometric representation of information based on multivoxel neural patterns across individuals, rather than simply linking a single behavioral measure with averaged activities across voxels in a certain region (*Kriegeskorte et al., 2008*; *Popal et al., 2019*). Here, with the help of IS-RSA, we were able to map differences in neural signals during bribery-related decision-making (i.e., Bribe vs. Control) directly onto key model-based parameters that characterize different types of moral costs specifically involved during bribery-related decision-making.

To address these questions, we designed a novel paradigm that captured the essence of a bribery situation while keeping it as simple as possible for the fMRI setting. Participants in the MRI scanner played the role of a power-holder and decided whether a fictitious proposer would earn a given amount of money or not (see *Figure 1*). The proposer was either alone or paired with a fictitious third party (scenario: Solo vs. Dyad). Critically, the proposer could obtain more profit by either lying or telling the truth (experimental condition: Bribe vs. Control; see below). To achieve this, the proposer offered an amount of money from this larger profit to bribe the power-holder. The task for the participants was to decide whether to accept or reject offers. Accepting the offer would profit both the power-holder and the proposer, whereas rejection would earn nothing for either of them. In the Dyad scenario, accepting the offer in the Bribe condition additionally harmed the interest of an innocent third party. Here, bribery is operationally defined as accepting an offer from a proposer who is fraudulently making a proposition that is more beneficial than that which should be honestly reported. Thus, the proposer who commits fraud for a higher personal profit can be regarded as a briber. Notably, this design allowed us to distinguish two types of moral costs that could be incurred by participants, as power-holders, when accepting bribes, namely conniving with a fraud committed by the briber and harming a third party. The former occurs in both scenarios, whereas the latter occurs only in the Dyad scenario.

## Results

### Behavioral and modeling results

#### Power-holders accepted less offers during bribery-related decision-making especially when it harmed a third party

We first investigated how participants' choice behaviors varied depending on the scenario (Solo/Dyad), and the proposer's conduct (Bribe/Control), after controlling for the effect of the offer proportion and that of the larger gain the proposer would earn in the reported option. Mixed-effect logistic analyses showed that participants were less likely to accept an offer in the Dyad scenario (main effect of the scenario: $\chi^2(1)=8.53$, p=0.003) and in the Bribe condition (main effect of the proposer's conduct: $\chi^2(1)=25.3$, p<0.001). More importantly, we observed a significant two-way interaction with respect to whether the offer was accepted ($\chi^2(1)=4.73$, p=0.030; see *Figure 2A, Figure 2—figure supplement 1*). Simple-effect analyses further revealed that participants were much less likely to accept bribes (vs. offers in the Control condition), in the Dyad scenario, where a third party was involved (acceptance rate: 61.6 ± 22.6% vs. 82.8 ± 13.2%; odds ratio = 0.15, $b = -1.91$ [–2.83, –1.25], *SE* = 0.41, p<0.001), in comparison to the Solo scenario (acceptance rate:

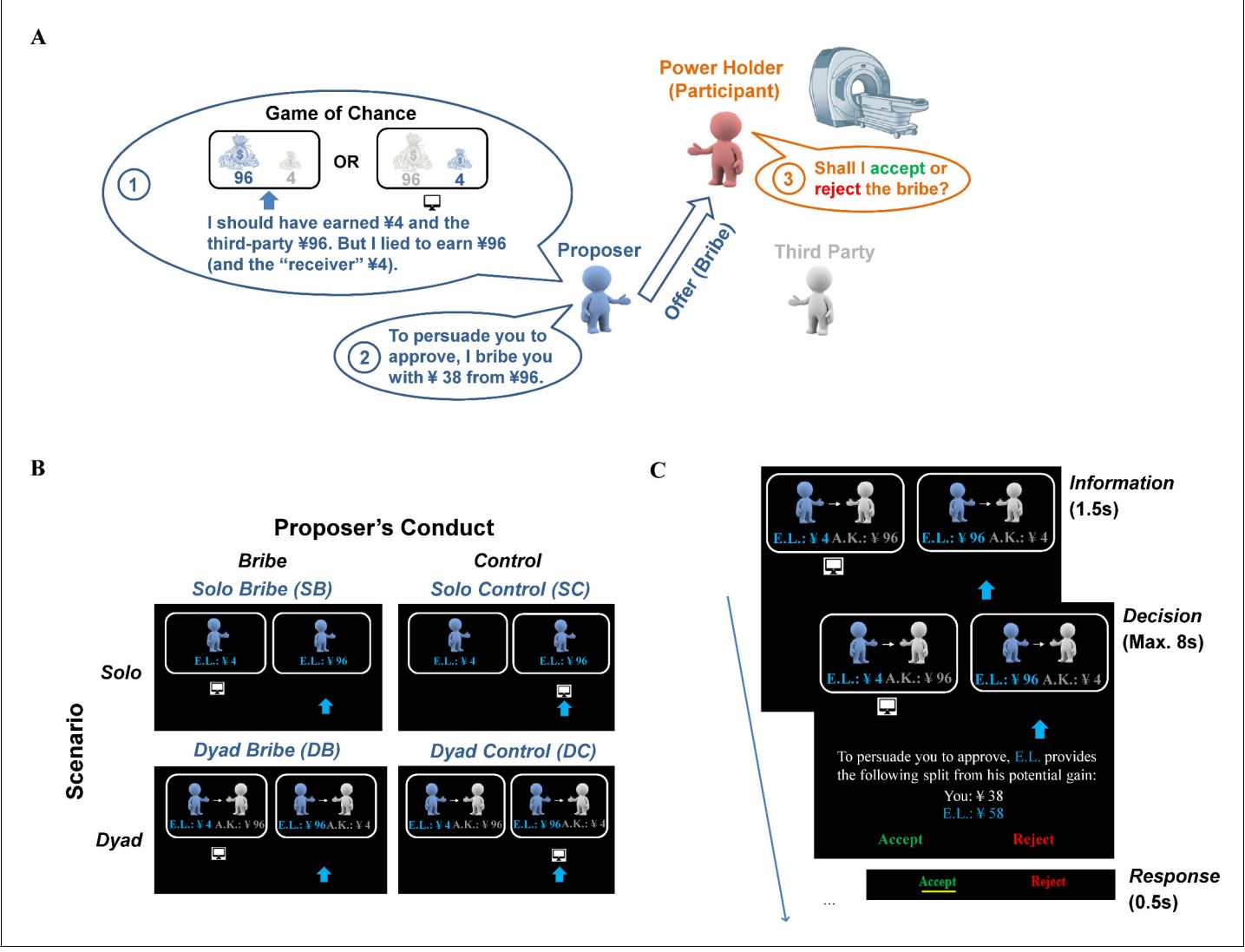

**Figure 1.** Task and design. (**A**) Schematic illustration of the behavioral paradigm. Here, we show the bribe case in the Dyad scenario (i.e., the Dyad Bribe condition [DB]). This condition consists of three roles: a proposer, a third party, and a power-holder (the real participant). Participants were endowed with the power to decide whether the proposer would earn a higher profit by lying, which also caused financial losses to a third party in a previous online study. The proposer hence bribes the power-holder, whose task was to decide whether to take the bribe. Notably, all proposers and third parties were framed as participants in a previous online study ('Game of Chance'), which was actually fictitious. (**B**) Illustration of all experimental conditions. We manipulated two factors, that is, scenario (Solo or Dyad) and proposer's conduct (Bribe or Control), which yielded four experimental conditions. (**C**) Trial procedure in the DB condition. In this example, a proposer (E.L.) lied by reporting the non-selected payoff option, which additionally harmed the interest of an innocent third party (A.K.), and bribed the power-holder with a certain amount of money from his/her potential gain (i.e., 38 out of 96 CNY). The participant needed to decide whether to accept or reject the bribe within 8 s (s). Once the decision was made (i.e., accepting the bribe here), a yellow bar appeared on the corresponding option to highlight the choice for 0.5 s, which was followed by a jittered fixation (i.e., 3–7 s with a mean of 5 s).

67.0 ± 20.7% vs. 83.3 ± 14.8%; odds ratio = 0.13, $b$ = −2.03 [−2.77, −1.08], $SE$ = 0.39, p<0.001; see *Supplementary file 1a* for details of the regression output).

Paralleling these findings, we found similar results with respect to the post-task subjective rating on the degree of the moral inappropriateness and the fraud of the proposer's behaviors (in the online game), and on the degree of the moral conflict during decision-making and the moral inappropriateness of their own behaviors. A within-subject multivariate analysis of variance (MANOVA) test showed a main effect of scenario ($F_{(1, 38)}$=5.59, p=0.001) and the proposer's conduct ($F_{(1, 38)}$=85.25, p=0.001) on these ratings. We also observed a significant two-way interaction effect ($F_{(1, 38)}$=4.87, p=0.003; see *Figure 2—figure supplement 1*), which was mainly driven by higher ratings

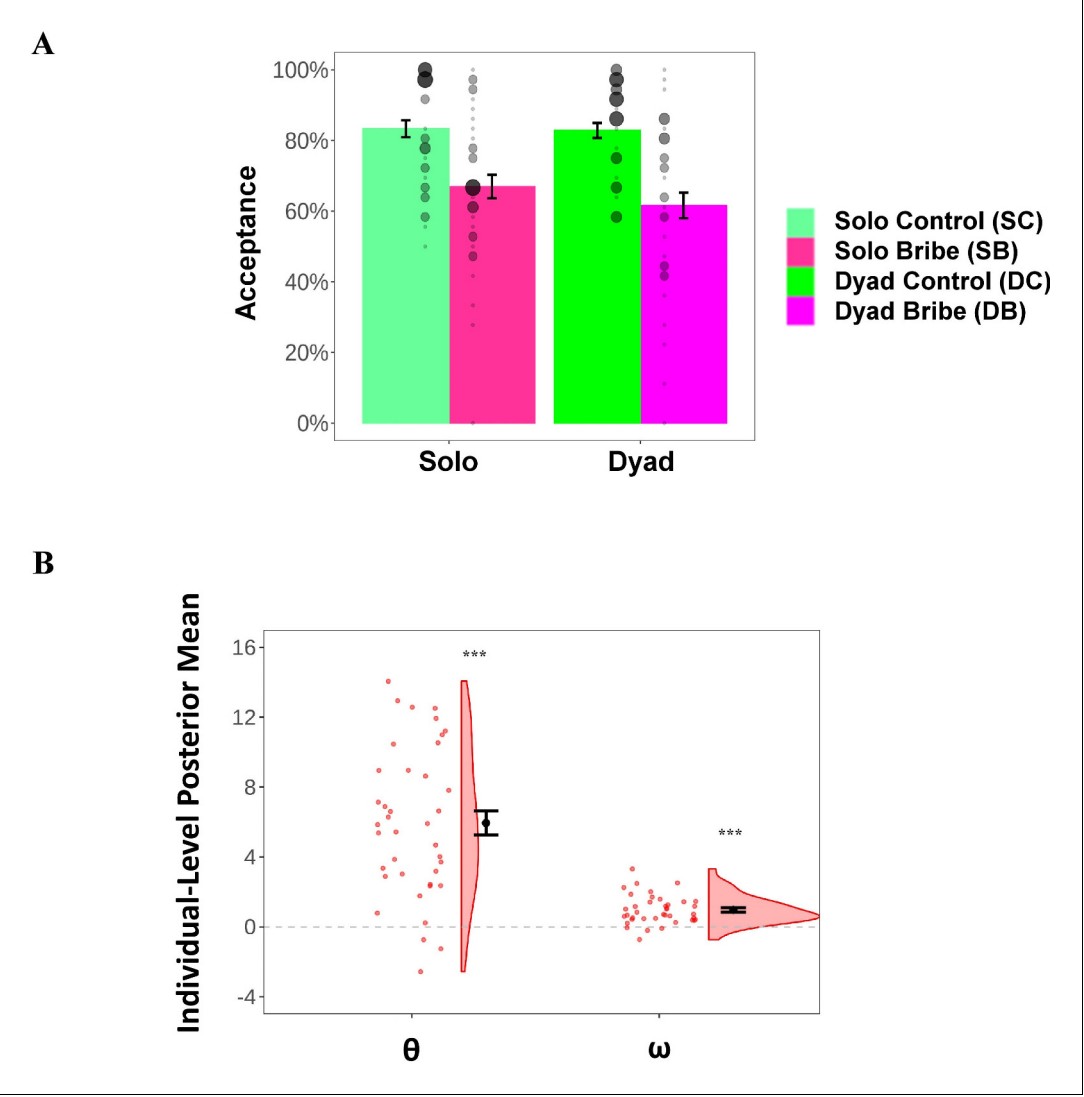

**Figure 2.** Results of behaviors and computational modeling. (**A**) Acceptance rate (%) as a function of context (Solo/Dyad) and the proposer's conduct (Control/Bribe). The significance was not marked because we did not perform statistical tests directly on these two dependent variables. Individual acceptance rates in each experimental condition are denoted by the black dots; the size of the dots represents the relevant sample size. (**B**) Posterior mean of individual-level key parameters characterizing moral costs (θ and ω) based on the winning model. Each black filled dot represents the group-level mean; each red dot represents the data of a single participant; the density curve represents the distribution of the data across all participants. Error bars represent the SEM; Significance: ***$p<0.001$.

The online version of this article includes the following figure supplement(s) for figure 2:

**Figure supplement 1.** Results of acceptance rate and subjective ratings.

**Figure supplement 2.** Results of parameter recovery analysis for each parameter in the winning model.

**Figure supplement 3.** Posterior predictive check of the winning model.

**Figure supplement 4.** Cross-validated posterior predictive check of the winning model.

**Figure supplement 5.** Results of reaction time (in ms).

in the Bribe (vs. Control) condition in the Dyad (*F*(1, 38)=124.43, $p<0.001$) in comparison to the Solo scenario (*F*(1, 38)=50.05, $p<0.001$).

## Power-holders took into account the moral costs during bribery-related decision-making

Next, we performed a model-based analysis on choice to understand the computational mechanisms underlying decisions of whether to take or refuse a bribe. To this end, we tested and compared a total of seven models with different utility functions (see Materials and methods for details). Parameters were estimated using the hierarchical Bayesian approach (HBA) via the 'hBayesDM' package. R-hat values of all estimated parameters of all models were smaller than 1.1 (the maximum R-hat value: 1.05), indicating adequate convergence of the Markov Chain Monte Carlo (MCMC) chains (*Gelman and Rubin, 1992*).

Based on hierarchical Bayesian model comparison (see *Supplementary file 1b*), we identified Model 5 (*Equation 1*) as the winning model, which was defined as below:

$$SV(p_P, p_T, p_{PH}) = \begin{cases} \beta_P p_P + (\beta_{PH} - \theta q)p_{PH} + \gamma|p_P - p_{PH}| & \text{if Solo scenario} \\ \beta_P p_P + \omega q p_T + (\beta_{PH} - \theta q)p_{PH} + \gamma|p_P - p_{PH}| & \text{if Dyad scenario} \end{cases} \tag{1}$$

where, in a given trial, SV denotes the subjective value, $p_P$, $p_{PH}$, $p_T$ represents the payoff (i.e., monetary gain) for the proposer ($_P$), the power-holder ($_{PH}$; participants), and the third party ($_T$) given different choices (i.e., accept or reject the offer), q is an indicator reflecting whether the trial tool place in the Bribe (q = 1) or the Control (q = 0) condition. $\beta_P$ and $\beta_{PH}$ are two independent free parameters capturing the weights on the payoff of the proposer and the power-holder, respectively. θ describes the moral cost brought by conniving with a fraud committed by a proposer. ω captures the moral cost brought by harming the interest of a third party. We also incorporated γ to account for the absolute payoff inequity between the proposer and the power-holder denoted in the offer (parameters range: $-20 \leq \beta_P, \beta_{PH}, \theta, \omega, \gamma <= 20$). Parameter recovery analysis showed that these true estimates (i.e., values we obtained from the actual choice data that we collected) positively correlated with the recovered estimates (i.e., what we obtained from the simulated data generated with these true parameters) for all parameters across individuals involved in the winning model (Pearson correlation: $rs > 0.71$, $ps <0.001$; see *Figure 2—figure supplement 2*; see Materials and methods for details), which suggests that our model is robustly identifiable. The posterior predictive check (PPC) further revealed that the proportion of acceptance predicted by this model successfully recovered the actual proportion of acceptance (all conditions: the within-sample PPC: $rs > 0.90$, $ps <0.001$; the out-of-sample PPC: $rs > 0.68$, $ps <0.001$; see *Figure 2—figure supplement 3*, *Figure 2—figure supplement 4*, and Materials and methods for details), which strengthened the results of the model comparisons.

Further analyses of the individual-level posterior mean of these parameters revealed that participants weighed the payoffs for themselves positively ($\beta_{PH}$: 16.02 ± 2.47, $t(38) = 40.51$, p<0.001, Cohen's $d = 6.49$) but did not do so for the proposer ($\beta_P$: −0.88 ± 2.48, $t(38) = −2.21$, p=0.033, Cohen's $d = −0.35$). Moreover, participants decreased the weights of their own payoffs when they were being bribed (θ: 5.94 ± 4.28, $t(38) = 8.67$, p<0.001, Cohen's $d = 1.39$) and apparently displayed concern for the financial loss of the innocent third party (ω: 0.97 ± 0.82, $t(38) = 7.32$, p<0.001, Cohen's $d = 1.17$; see *Figure 2B*; also see *Supplementary file 1c* for the bivariate correlation between all parameters in the winning model). These findings suggest that participants considered both forms of moral costs when deciding on whether to accept bribes. In addition, participants also disliked the inequality in the absolute payoff between themselves and the proposer (γ: −2.35 ± 2.23, $t(38) = −6.59$, p<0.001, Cohen's $d = −1.06$).

## Power-holders responded more slowly when accepting a bribe

We also investigated how participants' reaction time (RT) was modulated by the scenario and the proposer's conduct depending on the specific decisions (accept/reject), after controlling for the effect of the offer proportion and that of the larger gain the proposer would earn in the reported option. Before the analyses, we did a log-transformation on the RT due to its non-normal distribution (i.e., Anderson–Darling normality test: A = 232.54, p<0.001). Mixed-effect linear regression on log-transformed RT) showed that participants responded slower in the Dyad scenario (main effect of the scenario: $F(1, 5567)=15.52$, p<0.001), in the Bribe condition (main effect of the proposer's conduct: $F(1, 5570)=64.28$, p<0.001), and when they rejected the offer (main effect of decision: $F(1, 5582) =50.08$, p<0.001). Moreover, we detected a two-way interaction between the decision (accept/

reject) and the proposer's conduct (bribe/control; $F(1, 5575)=20.79$, p<0.001). *Post-hoc* analyses further showed that participants responded more slowly when they accepted offers in the Dyad scenario (vs. Solo; $b = 0.07$ (0.03, 0.11), $SE = 0.02$, $t(4093) = 3.724$, p<0.001, Cohen's $d = 0.12$) or when the proposer was bribing them (vs. Control; $b = 0.20$ (0.16, 0.24), $SE = 0.02$, $t(4095) = 9.990$, p<0.001, Cohen's $d = 0.31$). Neither of these main effects was identified for rejection decisions (Dyad vs. Solo: $b = 0.02$ (−0.06, 0.10), $SE = 0.04$, $t(1435) = 0.561$, p=0.575, Cohen's $d = 0.03$; Bribe vs. Control: $b = −0.01$ (-0.08, 0.06), $SE = 0.04$, $t(1439) = −0.269$, p=0.788, Cohen's $d = −0.01$; see *Figure 2—figure supplement 5* and *Supplementary file 1d* for the descriptive summary of the original RT; see *Supplementary file 1e* for details of the regression output.

## Neuroimaging results

### vAI represents the effect of the moral cost of conniving with fraud on expected personal profits

We implemented the general linear model (GLM) analyses to test specific hypotheses concerning different research questions (see Materials and methods for details of GLM analyses). Notably, all a-priori hypotheses and planned analyses were not pre-registered. We first examined brain regions encoding the moral cost of conniving with a briber on the expected gains for both the proposer and the participant due to bribe-taking (as parametric modulators, PM; GLM1a), with a focus on vAI. Consistent with our hypothesis, we observed stronger engagement of the left vAI in the Bribe (vs. Control) condition regardless of scenarios (peak MNI coordinates: −40/10/−6, $t(114) = 3.81$, $p$(SVC-FWE)=0.025) when investigating brain regions encoding the expected personal profits (i.e., the profits earned by participants had they accepted the offer). To examine the lateralization of this effect, we lowered the statistical threshold and found that the right vAI delineated similar neural responses (peak MNI coordinates: 36/12/–14, $t(114) = 3.27$, $p$(SVC-FWE)=0.090; see *Figure 3A*). We also found that the right vAI displayed an increased sensitivity to the expected personal gains in the Dyad (vs. Solo) scenario (peak MNI: 36/14/–14; $t(114) = 3.86$, p(SVC-FWE)=0.022), whereas the left vAI showed a similar trend that did not reach statistical significance (peak MNI: −36/10/−16; $t(114) = 3.09$, p(SVC-FWE)=0.146; see *Supplementary file 1f* for results of other contrasts under a lenient threshold). In addition, we also observed a stronger activity in the dorsal part of anterior cingulate cortex (dACC) during bribery-related decision-making (i.e., event contrast: Bribe vs. Control in GLM1c; same below) across scenarios, under a whole-brain threshold (see *Figure 3—figure supplement 1*). No region was observed in other contrasts (see *Supplementary file 1f* for other regions activated under a lenient threshold).

### TPJ encodes the expected losses for the third party and distinguishes bribery-related decision-making in the two scenarios

We then examined whether TPJ was involved in encoding the expected losses incurred by the innocent third party due to bribe-taking (as the only PM in the Dyad Bribe [DB] condition; GLM1b). As predicted, we observed a strong involvement of the right TPJ, extending to the posterior part of the superior temporal sulcus, in encoding this PM (pSTS; peak MNI coordinates: 54/–48/–4, $t(38) = 4.70$, $p$(SVC-FWE)=0.010; see *Figure 3B*; also see *Supplementary file 1f* for other activated regions). Results still held after controlling the effect of expected gains for both the proposer and the participant (peak MNI: 54/–48/–4; $t(38) = 4.63$, p(SVC-FWE)=0.012; see *Figure 3—figure supplement 2*).

This finding suggested another possibility that TPJ/pSTS might be selectively engaged during bribery-related decision-making in the Dyad (vs. Solo) scenario, where the interest of a third party is explicitly compromised. To test this hypothesis, we first performed *post-hoc* univariate analyses but did not find any supportive evidence (i.e., no significant TPJ/pSTS signal was observed in the following contrasts of GLM1c; contrast 1: DB vs. Dyad Control [DC]; contrast 2: (DB – DC) vs. (Solo Bribe [SB] – Solo Control [SC])). However, using a *post-hoc* multivariate-based decoding analysis with a leave-one-subject-out (LOSO) cross-validated approach, we found that the decision-relevant multivoxel activation patterns of TPJ/pSTS only dissociated the bribery-related decision-making in the Dyad (DB vs. DC; two-choice forced-alternative accuracy ± SE: 69.2 ± 11.1%, p=0.024; $p_{\text{permutation}}$ = 0.020) but not the Solo scenario (SB vs. SC; 53.8 ± 8.6%, p=0.749; $p_{\text{permutation}}$ = 0.300; see *Figure 3C*; also see *Figure 3—figure supplement 3* for decoding results of the left and right TPJ/

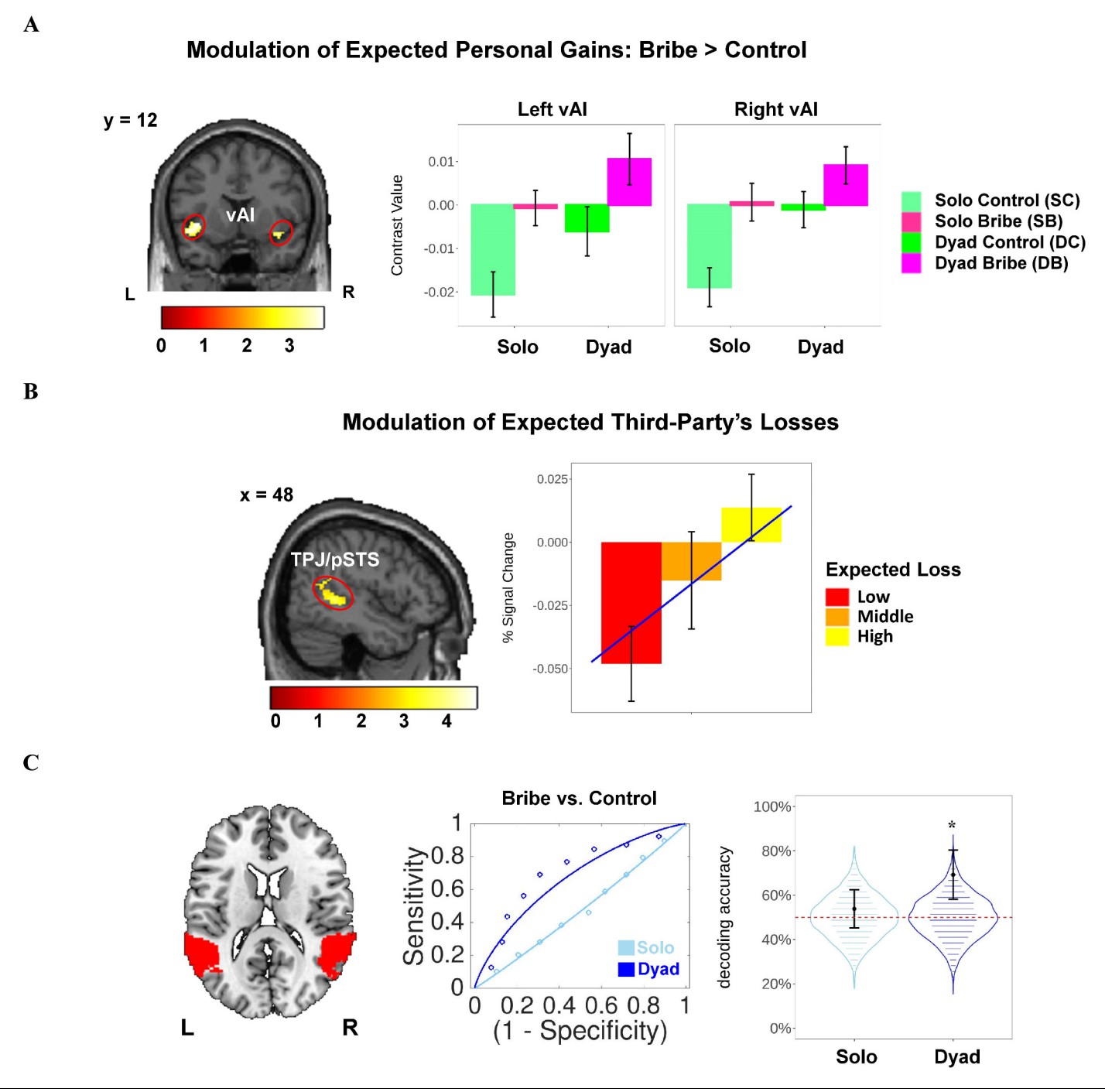

**Figure 3.** Parametric modulation of expected personal gains and potential loss for the third-party during bribery-related decision-making. (**A**) Enhanced ventral part of anterior insula (vAI) signal by the modulation of the expected personal gain during bribery-related decision-making (Bribe vs. Control; GLM1a). The contrast values of the activated cluster were extracted for visualization. (**B**) The parametric modulation of the expected potential loss for the third party in the temporoparietal junction/posterior superior temporal sulcus (TPJ/pSTS) (GLM1b). Notably, this parametric modulation only exists in the Dyad Bribe condition due to the experimental design. To visualize the effect of parametric modulation, we split these continuous parameters into three bins (low: 0–33%; medium: 33–67%; and high: 67–100%), re-estimated the general linear model and extracted the percent signal change (PSC) in relevant activated clusters via rfxplot (http://rfxplot.sourceforge.net/; *Gläscher, 2009*). Each line refers to the linear fit between the bin and the PSC. (**C**) Multivariate patterns of TPJ/pSTS distinguished bribe (vs. Control) only in the Dyad scenario. The regions of interest (ROI) of TPJ/pSTS was defined based on a whole-brain parcellation map from the Neurosynth database (left). The receiver operating characteristic curve showed that the multivoxel TPJ/pSTS activity pattern during decision-making can distinguish the Bribe condition from the Control condition only in the Dyad scenario but not in

*Figure 3 continued on next page*

*Figure 3 continued*

the Solo scenario (middle). Permutation test (N = 5000) further illustrated that the two-choice forced alternatives decoding accuracy in TPJ/pSTS in the Dyad scenario were unlikely achieved by chance (p=0.020; right). The violin plot (right) describes the null distribution of decoding accuracy of the multivoxel TPJ patterns during decision-making that distinguishes the bribe from the Control condition in the Solo and Dyad scenarios, respectively, obtained via 5000 times of permutation. Significance: *p<0.05. Error bars refer to SEM. Display threshold: voxel-level p (uncorrected) <0.005 (**A, B**). L: left; R: right.

The online version of this article includes the following figure supplement(s) for figure 3:

**Figure supplement 1.** Enhanced dorsal part of anterior cingulate cortex (dACC) activity during bribery-related decision-making (Bribe vs. Control; GLM1c).

**Figure supplement 2.** Parametric modulation of the expected loss to the third party in the temporoparietal junction/posterior superior temporal sulcus (TPJ/pSTS) after controlling for the effect of expected gains either for the proposer or the participant (GLM1b-s).

**Figure supplement 3.** Results of decoding analyses using decision-related multivoxel neural patterns of (**A**) the left and (**B**) the right temporoparietal junction/posterior superior temporal sulcus (TPJ/pSTS).

pSTS). This finding corroborated what we found in the parametric analysis and further indicated that TPJ/pSTS works differently between the Solo and the Dyad scenarios by showing distinct activation patterns rather than a mean activation intensity across voxels, during bribery-related decision-making.

## vmPFC encodes relative SV during bribery-related decision-making

Next, we investigated the brain regions encoding the relative SV during bribery-related decision-making. To this end, we established GLM2a to distinguish the Bribe condition from the Control condition (by pooling the Solo and Dyad scenarios). The relative SV was defined by subtracting the SV of the non-chosen option from that of the chosen option (i.e., relative $SV = SV_{chosen} - SV_{unchosen}$) based on the winning model. As expected, the vmPFC was more engaged in integrating the value signals in the Bribe condition (peak MNI coordinates: 0/34/−10, $t(38) = 4.73$, $p$(SVC-FWE)=0.007). A smaller effect was also observed in the Control condition (peak MNI coordinates: 0/52/−10, $t(38) = 3.76$, $p$(SVC-FWE)=0.064; see *Figure 4A*). These findings suggest that the vmPFC played a common role in computing an integrated value signal during the decision period regardless of whether the decision-making concerned bribery. To further test this hypothesis, we performed additional analyses that provided supportive evidence.

First, a direct comparison between the value signal of vmPFC in the Bribe and the Control conditions did not reveal any significant difference. A supplementary pattern similarity analysis revealed a significantly positive correlation between multivoxel vmPFC patterns of value computation in the Bribe and the Control conditions ($r = 0.14 \pm 0.36$, $p$(permutation)=0.023; see *Figure 4B, C*). In addition, we also ran a separate GLM (GLM 2b) that pooled all conditions as a single regressor to investigate the neural network generally involved in value computation. We found that vmPFC, together with bilateral middle/inferior temporal gyri and posterior cingulate cortex extending to cuneus, correlated positively with the relative SV signal (see *Figure 4—figure supplement 1*; also see *Supplementary file 1g* for other activated regions).

## dlPFC is selectively engaged in rejecting bribes in the Dyad scenario

We also examined how neural signals during bribery-related decision-making were associated with different choices and how they varied depending on scenarios. To this end, we established GLM3 that contained the onset of each condition with respect to specific choices (i.e., eight onset regressors in total, namely *accept* or *reject* in conditions of SC, SB, DC, and DB). To simplify the analysis, we computed the neural activity specific to rejecting as opposed to accepting offers (i.e., reject vs. accept) in all four conditions and then defined the anti-corruption neural signals with such rejection-specific neural activity in the Bribe condition (i.e., contrast: Bribe$_{(reject - accept)}$ − Control$_{(reject - accept)}$). Given our hypothesis, we focused on the dlPFC in this analysis. We found an increased anti-corruption signal in the dlPFC in the Dyad scenario compared with the Solo scenario (left dlPFC: peak MNI coordinates: −38/48/18, $t(96) = 3.70$, $p$(SVC-FWE)=0.024; right dlPFC: peak MNI coordinates: 42/50/16, $t(96) = 3.52$, $p$(SVC-FWE)=0.050; see *Supplementary file 1h* for other activated regions). Importantly, *post-hoc* analyses revealed that the anti-corruption signal in the dlPFC was significantly

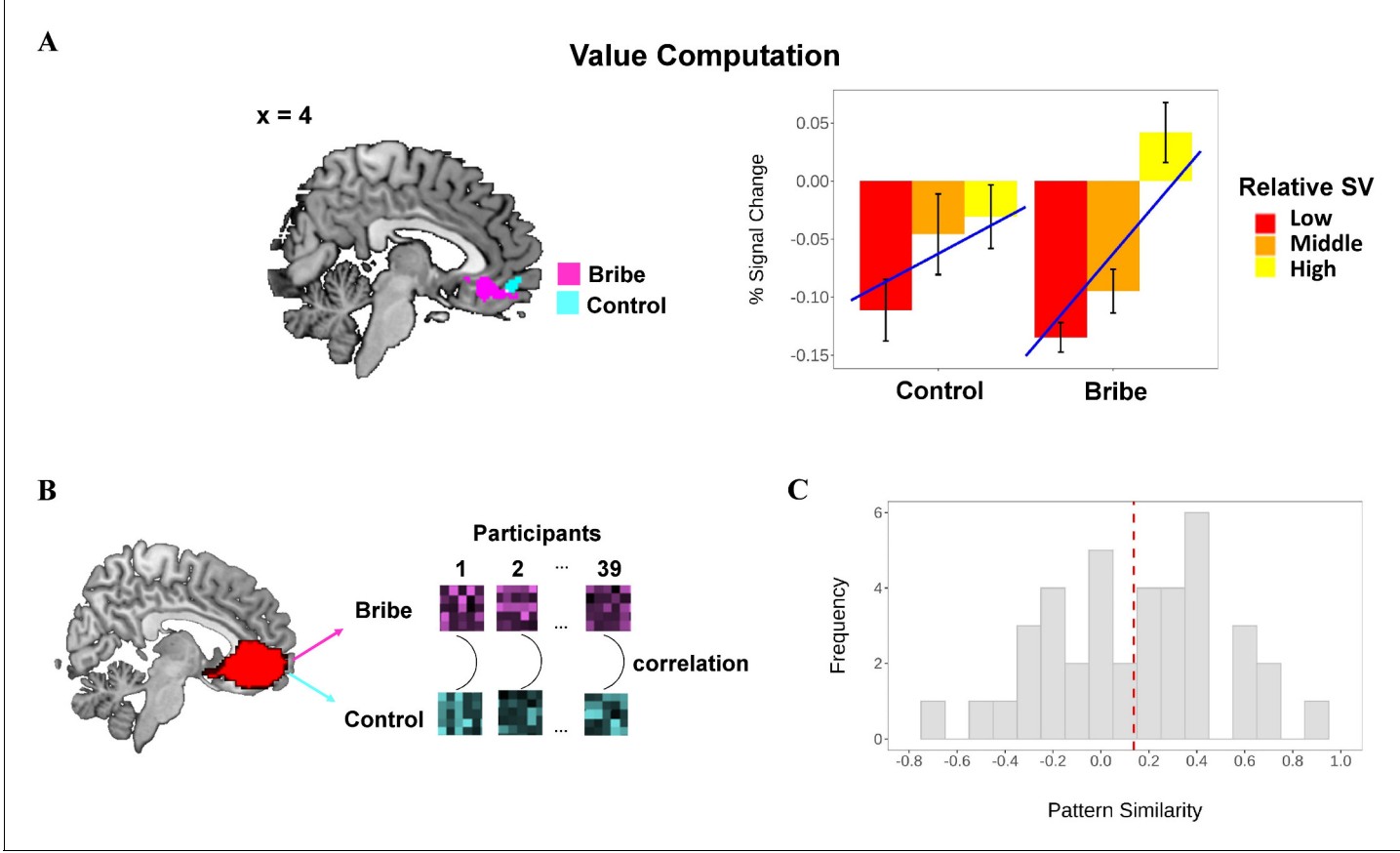

**Figure 4.** Value computation in the ventromedial prefrontal cortex (vmPFC) during bribery-related decision-making. (**A**) Ventromedial prefrontal cortex (vmPFC) encodes relative subjective value (SV) in the Bribe and the Control conditions, respectively. Relative SV was defined as the SV difference between the chosen and the non-chosen option. To visualize the effect of parametric modulation, we split the relative SV into three bins, that is, low (0–33%), medium (33–67%), and high (67–100%), re-estimated the general linear model and extracted the percent signal change (PSC) in the activated cluster via rfxplot (http://rfxplot.sourceforge.net/). The line refers to the linear fit between the bin and the PSC. (**B**) Procedure of pattern similarity analyses. For each participant, we extracted the multivoxel neural patterns (i.e., those heat maps; only for illustration) within the vmPFC mask from the parametric contrast of relative SV in Bribe and Control conditions, respectively (pooling the Solo and Dyad scenarios). Next, we computed the similarity (correlation) between these neural patterns in the two conditions. For statistical analysis, all correlation coefficients were transformed to Fisher's z value. (**C**) Histogram of the distribution of pattern similarity across all participants. The vertical dashed line refers to the mean of the pattern similarity. The online version of this article includes the following figure supplement(s) for figure 4:

**Figure supplement 1.** Brain regions positively correlate relative subjective value (SV) during decision period regardless of experimental conditions (GLM2b).

higher than 0 only in the Dyad scenario (ps <0.002), but not the Solo scenario (ps >0.056; see *Figure 5A, Figure 5—figure supplement 1*).

## Individual differences in susceptibility to the moral cost of conniving with fraud modulates vmPFC–dlPFC functional connectivity during bribery-related decision-making

Using a general psycho-physiological interaction (gPPI) analysis, we further tested whether there was functional connectivity between the vmPFC and dlPFC (and other brain regions) during bribery-related decision-making. No region was detected that significantly enhanced the functional connectivity with vmPFC during bribery-related decision-making in general. However, we observed a negative correlation between the functional coupling of the vmPFC and the right dlPFC during bribery-related decision-making (i.e., PPI contrast: Bribe vs. Control) and the parameter θ(peak MNI coordinates: 46/36/30, $t(37)$ = 4.78, $p$(SVC-FWE)=0.005; see *Figure 5B*). This parameter characterizes the aversion to conniving with the fraud committed by the briber. The control analysis confirmed the

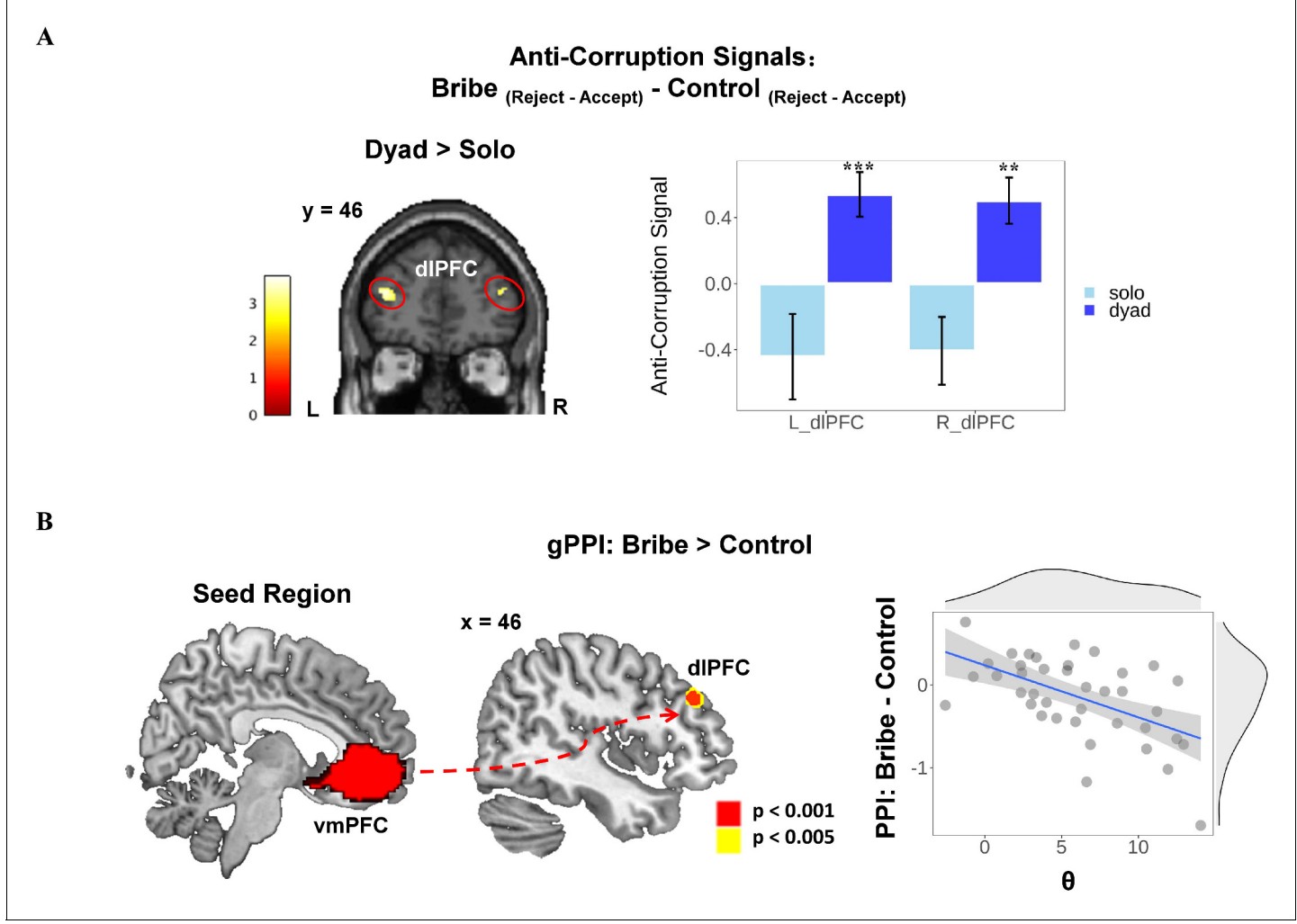

**Figure 5.** Context-dependent anti-corruption signal and the functional connectivity results. (**A**) Anti-corruption signal in dorsolateral prefrontal cortex (dlPFC) was specifically enhanced in the Dyad scenario. We defined the anti-corruption signal as the activities (i.e., contrast value) of rejection (vs. acceptance) choice that were specific to the Bribe (vs. Control) condition. Post-hoc analyses on anti-corruption signals within the activated clusters of bilateral dlPFC showed that the interaction was mainly driven by increased anti-corruption signals in the Dyad scenario. Display threshold: voxel-level p (uncorrected) <0.001. Error bars represent the SEM; significance: ***p<0.001. (**B**) Individual differences in θ negatively modulate the functional connectivity between ventromedial prefrontal cortex (vmPFC) and right dlPFC during bribery-related decision-making (i.e., Bribe vs. Control). The seed region of vmPFC was defined based on a whole-brain parcellation given a meta-analytic functional coactivation map of the Neurosynth database (http://neurovault.org/images/39711/). To visualize the relationship, we extracted the contrast value (cv) of the psycho-physiological interaction regressors of the Bribe and Control conditions within the activated cluster and plotted the correlation between the cv and the individual parameter. Each gray dot represents the data of a single participant; the blue line represents the linear fit using robust correlation (r = −0.54, 95% CI: −0.88,–0.19, p=0.003); the gray density curve indicates the distribution of each variable, respectively.

The online version of this article includes the following figure supplement(s) for figure 5:

**Figure supplement 1.** Modulation of scenario on the dorsolateral prefrontal cortex (dlPFC) activity during bribery-related decision-making regarding specific choice.

above results in the right dlPFC after ruling out the effect of parameter ω (peak MNI coordinates: 46/36/30, *t*(36) = 4.58, *p*(SVC-FWE)=0.014; see *Supplementary file 1i* for other activated regions). The left dlPFC also showed a similar connectivity pattern with vmPFC when we adopted a lenient threshold (peak MNI coordinates: 42/44/18, *t*(37) = 3.40, *p*(SVC-FWE)=0.093; after controlling for the effect of parameter ω: peak MNI coordinates: 42/44/18, *t*(37) = 3.58, *p*(SVC-FWE)=0.065).

## Individual differences in bribery-related dlPFC activity represent variations in bribery-specific preference

Finally, we investigated how inter-individual differences in bribery-specific preferences were implemented in the brain. In particular, we focused on the dlPFC because decision-related neural signals in this region have been shown to be modulated by inter-individual variations of moral behaviors (e. g., self-serving dishonesty; *Yin and Weber, 2018*) and other-regarding preferences in moral decision-making (e.g., harm aversion; *Crockett et al., 2017*).

To this end, we used an IS-RSA, which allowed us to directly investigate whether the neural patterns in dlPFC during bribery-related decision-making are similar for participants who displayed similar bribery-specific preferences for bribery, measured by the moral costs involved during bribe-related decision-making (i.e., θ and ω). In particular, we established a parameter representational dissimilarity matrix (RDM) by calculating the Euclidean distance between the two parameters specifically characterizing the moral costs (i.e., θ and ω; the correlation between these two parameters: $r$ (37) = 0.258, p=0.113). We next constructed the neural RDMs by calculating the correlation dissimilarity of the parameter estimates of subjective-level contrasts during bribery-related decision-making extracted from the bilateral dlPFC (i.e., the contrast of Bribe vs. Control). We then measured the

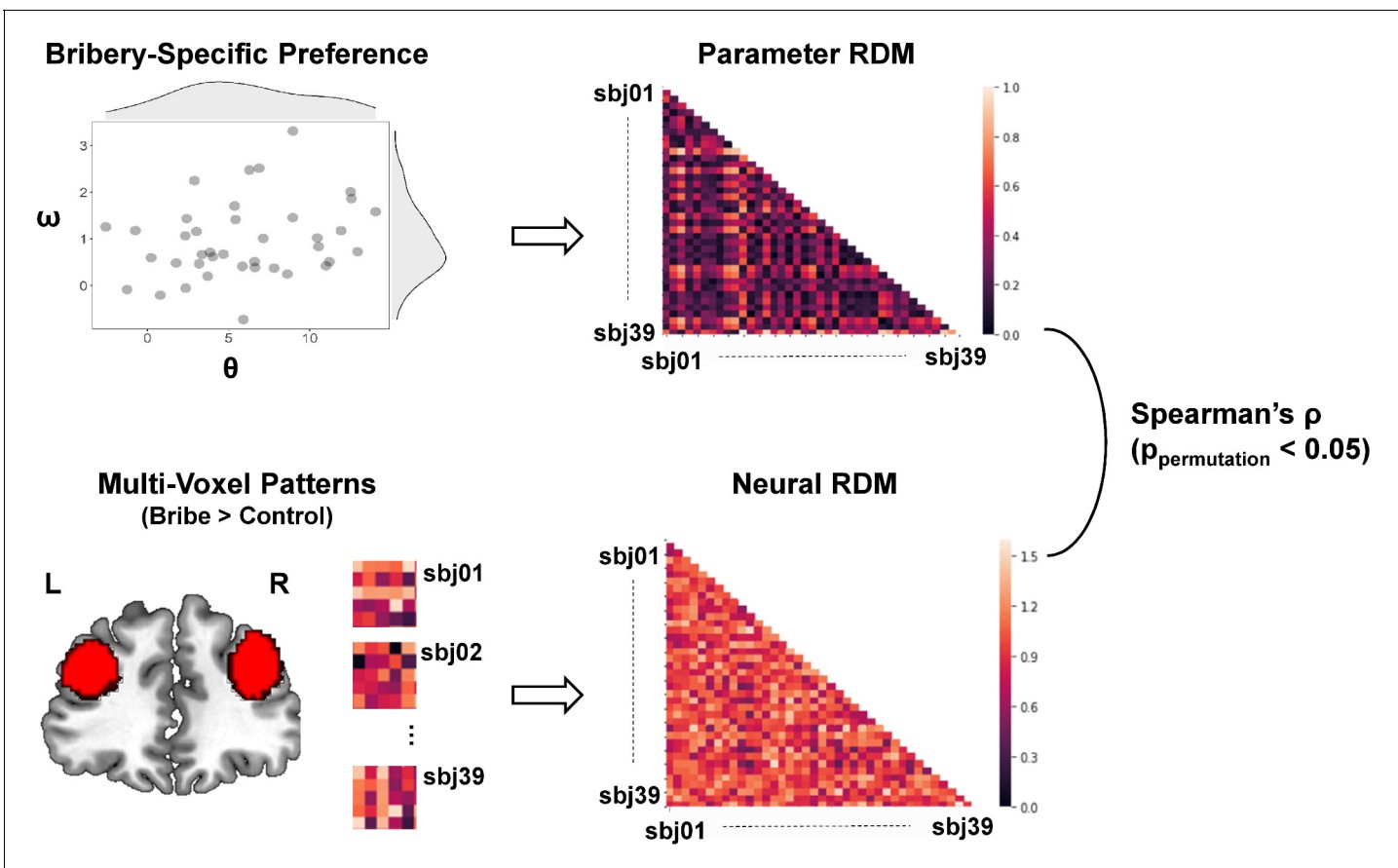

**Figure 6.** Illustration of the inter-subject representational similarity analysis. We created a parameter representational dissimilarity matrix (RDM), which measures the dissimilarity across participants in the bribery-specific preference that is calculated by the Euclidean distance between each pair of participants in θ and ω driven from the winning model. Both parameters together characterize the individual differences in bribery-specific preference (see the scatter plot: each dot represents the data of a single participant; the density curve indicates the distribution of each parameter, respectively). We also created a neural RDM, which measures the dissimilarity across participants in the neural activities within bilateral dorsolateral prefrontal cortex (dlPFC) during bribery-related decision-making (i.e., Bribe vs. Control) that is measured by the correlation distance between the multivoxel patterns of each pair of participants (i.e., those heat maps; only for illustration). The region of dlPFC was defined based on a whole-brain parcellation given a meta-analytic functional coactivation map of the Neurosynth database (http://neurovault.org/images/39711/). We then calculated the Spearman rank-order correlation between these two RDMs and implemented a permutation test to confirm the statistical significance.

similarity between the neural RDMs and the preference RDM (see *Figure 6*). The results revealed a significant inter-subject similarity effect in the decision activity pattern, specific to corruption in the dlPFC (Spearman's *rho* = 0.108, $p_{permutation}$ = 0.019).

## Discussion

We have used a novel paradigm that captures the core components of real-life bribery to offer a behavioral and neural characterization of how a power-holder determines whether or not to accept bribes. Both the model-free behavioral results and the computational modeling findings indicate that when facing the temptation of a bribe power-holders take into account the moral costs of conniving with the briber (measured by the parameter θ) and the harm inflicted on a third party (measured by the parameter ω). Incorporating these moral costs into the SV computation explains why the probability of accepting a bribe is lower than that of accepting an otherwise-comparable honest offer. This is especially the case when accepting the bribe also harms a third party. Post-task subjective ratings also confirm that participants did feel more moral conflict while making decisions in the Bribe (vs. Control) condition. They also felt it was more morally inappropriate to accept bribes than honest offers, especially when doing so harmed a third party. Our study extends to the context of corruption previous behavioral results on dishonesty that reveal people are generally aversive to dishonesty (*López-Pérez and Spiegelman, 2013*) and that deception is less morally acceptable when it harms another's interests (*Lindskold and Walters, 1983*).

At the neural level, we explored the neurocomputational mechanisms underlying bribery-related decision-making using model-based fMRI. First, we identified the neural bases of two types of moral costs engaged when deciding whether to take a bribe (i.e., Bribe vs. Control). The vAI was more sensitive to the expected ill-gotten personal gains (PM contrast: Bribe vs. Control). Interestingly, we also observed that the vAI (especially the right part) is more engaged in encoding expected personal gains in the Dyad (vs. Solo) scenario. The vAI plays a critical role in guiding dishonest decisions under various circumstances (*Yin et al., 2017*) and in perceiving other's dishonest intentions (*Yin and Weber, 2016*). These findings can be broadly linked to the modulation of aversive feelings by vAI that generate motivation to social norm enforcement (*Bellucci et al., 2018*). Consistent with previous studies, our results show that a key computation performed by the vAI signal is to encode bribery-related profits, especially when a potential victim is involved in the social context. This signal might reflect an aversive feeling towards moral transgression due to bribe-taking behavior. This might contribute, but not necessarily lead, to preventing power-holders from being corrupted. Moreover, the right TPJ/pSTS shows the expected role in tracking the expected loss to a third party due to potential bribe-taking decisions in the Dyad scenario. It is well-established that TPJ/pSTS (especially the right part) functions as a core component of the mentalizing brain network (*Schurz et al., 2014*). Mentalizing ability is a prerequisite for making judgments and decisions that take into account the welfare of others (*de Waal, 2008*), mainly *via* recruitment of the right TPJ (*Young et al., 2010*). Indeed, recent neuroimaging studies have shown that TPJ/pSTS is critically involved in pitting personal interests against other-regarding (or moral) concerns during prosocial decision-making (*Morishima et al., 2012*; *Tusche et al., 2016*). Supporting this claim, we also found that the multivoxel neural patterns of TPJ extending to pSTS selectively discriminated bribery-related decision-making (i.e., Bribe vs. Control) only when a third party victim was explicitly involved. Both our findings indicate that TPJ/pSTS is likely to reflect the moral cost incurred by participants when harming a third party.

We also observed a strong involvement of dACC during bribery-related decision-making (event contrast: Bribe vs. Control). Together with the aforementioned results of brain regions encoding moral cost, these findings also build an interesting link to the literature of guilt. Several neuroimaging studies adopting different paradigms (*Bastin et al., 2016*; *Takahashi et al., 2004*; *Yu et al., 2014*) consistently report the engagement of dACC (adjacent to the mid-anterior cingulate or dorsomedial prefrontal areas), AI, and TPJ when guilt-specific feelings are elicited. Indeed, theoretical and behavioral evidence shows that either guilt proneness (as a personality trait) or (anticipated or elicited) guilt feelings might curb immoral behaviors including cheating or bribe-taking (*Balafoutas, 2011*; *Cohen et al., 2012*; *Köbis et al., 2016*; *Motro et al., 2016*). Although investigating the role of guilt in corrupt actions is not the goal of the current study, our results suggest a promising future direction of exploring the effect of social/moral emotions in bribery-related decision-making.

We further showed how value computation engaged in the trade-off between personal profits and bribe-related moral costs is implemented in the brain. As predicted, these moral costs along with monetary payoffs are integrated into a value signal computed in the vmPFC, together with other regions (e.g., posterior cingulate cortex). The vmPFC is known to be a central hub of the brain valuation network (*Bartra et al., 2013*). This region has been reported to be involved in guiding choices in various domains via the computation of the SV of the available options, and it is thought to encode the common neural currency for value (*Levy and Glimcher, 2012*). Our findings extend recent fMRI studies indicating that vmPFC is involved in the trade-off between moral values and monetary profits to the domain of corruption (*Crockett et al., 2017*; *Qu et al., 2020*).

Another critical contribution of this study is to address how specific choices, such as accepting or rejecting a bribe, shape the neural substrates of decision-making in various bribe-related scenarios. We found that a higher anti-corruption signal in the dlPFC, defined as the neural activity of rejecting (vs. accepting) an offer specifically in the Bribe (vs. Control) condition, was identified in the Dyad scenario but not the Solo scenario. This finding parallels our behavioral results showing that participants were less likely to accept a bribe and rated this behavior as the most morally inappropriate when the interests of a third party were compromised. Thus, these findings suggest that the dlPFC plays a dedicated role in ethical behavior to guide the choice of rejecting a bribe in a situation combining different forms of moral costs and is specifically sensitive to harm caused to third parties. Consistent with this claim, recent studies reported that the lateral prefrontal cortex is more engaged in encoding profits that were obtained by inflicting physical pain on others (*Crockett et al., 2017*) and when individuals flexibly align moral goals with value representations (*Carlson and Crockett, 2018*).

We also provide novel empirical evidence regarding how inter-individual differences in bribery-specific preferences modulate neural activities during bribery-related decision-making. On the one hand, the functional coupling between vmPFC and dlPFC during bribery-related decision-making (i. e., Bribe vs. Control), decreased with the inter-individual level of the moral cost of conniving with a briber (i.e., a fraudulent proposer; as reflected by the θ parameter). Previous literature has consistently shown an increased vmPFC–dlPFC functional connectivity when self-control is required during various types of value-based decision-making (*Hare et al., 2009*; *Hare et al., 2014*). This leads to our initial prediction that a generally stronger vmPFC–dlPFC functional connectivity would be observed during bribery-related decision-making (Bribe vs. Control). However, recent evidence also suggests that the strength of the vmPFC–dlPFC coupling, enhanced by self-control, might vary from person to person. For instance, a recent study has shown inter-individual differences in vmPFC–dlPFC connectivity when resisting to trade honesty values against economic benefits (*Dogan et al., 2016*). This indicates that the modulatory role of dlPFC on value representation is flexible, according to individuals' other-regarding preferences. In agreement, our result of functional connectivity suggests that more self-control might be needed to devalue the temptation to earn morally tainted profits for those power-holders who are less aversive to the briber's fraudulence (i.e., participants with smaller θ). On the other hand, using IS-RSA, we revealed that neural patterns during bribery-related decision-making in the dlPFC were similar for participants who displayed similar preferences for bribery (i.e., indexed by a dissimilarity matrix of θ and ω across participants). Standard univariate regression analyses have established that this region is linked to individual variations in concerns for dishonesty and related behaviors (*Dogan et al., 2016*; *Yin and Weber, 2018*), as well as in harm aversion (*Crockett et al., 2017*). Compared with the traditional univariate approaches, IS-RSA can directly map multidimensional psychological states (computations) between individuals to specific neural patterns (*Kriegeskorte et al., 2008*; *Popal et al., 2019*). Thus, the multivariate approach allows us to leverage both individual differences in task-based preferences of corruption and decision-relevant multivoxel neural patterns. This reveals that the dlPFC represents a complex geometry characterized by a multidimensional model space of moral preferences across individuals.

Notably, all these results provide evidence for a critical role of the dlPFC in different aspects of bribery-related decision-making. The univariate result (based on GLM3) suggests that, from a within-subject perspective, the dlPFC plays a dedicated role in guiding the choice to reject bribes in a context-dependent manner. Both the functional connectivity results and the IS-RSA results concern the between-subject perspective. In general, these findings indicate that the dlPFC signature (or pattern) might reflect the preference of whether to accept or refuse a bribe across individuals as power-holders. These results are consistent with a causal role of dlPFC in ethical behaviors (*Maréchal et al.,*

*2017*; *Zhu et al., 2014*) and may inspire future studies to investigate whether such a causal role extends to the corrupt behaviors and how it varies between individuals.

Several issues concerning the present study need further discussion. First, although the present task captures the essence of corruption from the perspective of a person in power, it only simulates a specific type of corruption in a lab setting. The real phenomenon is far more complex and diverse. This obviously constrains the generalization of the current findings to corruption in field settings. Second, the present task did not contain a condition in which the proposer honestly reported the option with the lower payoff having been randomly selected by the computer. We did this purposely to preserve the symmetry of the experimental design and maintain the motivation of proposers' behaviors to being entirely rational and explainable by the optimization of personal profits. Thus the offer proposition, in either the Bribe or the Control condition, was realistic for the proposer, that is, it would not make sense if the proposer proposed an offer to earn a lower payoff. In addition, adding such trials to the current task would inevitably prolong the duration of the experiment, which might make participants more tired and thus influence the quality of the data we collected. Third, the present task adopted a multiround single-shot economic game that did not involve any real partner physically present during the experiment. We decided to use such an experimental setting not only because it is commonly used in neuroeconomics studies (e.g., *Aimone et al., 2014*; *Hu et al., 2018*; *Spitzer et al., 2007*) but also because, here, each choice can be considered independent, thus alleviating the potentially confounding effects of learning and concerns of reputation in repeated interactions. Although the lack of social interactions in the task might diminish the involvement of participants and influence their beliefs about the authenticity of the experiment setting, our behavioral results were consistent with the predictions that were built on the assumption that participants believed the existence of proposers and third parties. Nevertheless, the concern raised above cannot be completely ruled out because participants were never asked about whether or not they believed the cover story. All these limitations, we believe, should be addressed in future studies.

To conclude, the present study provides a neurocomputational account characterizing how a power-holder reaches corrupt decisions by weighing bribe-related moral costs and material gains. It shows that corruption is controlled by an interconnected brain network with nodes processing specific computational signals. Moreover, our study offers new clues on how individuals who vary in bribery-specific preferences differ in neural signals during bribery-related decision-making. These findings open a new gate for improving our understanding of the neurobiological mechanisms underlying corruption from a micro-level using a multidisciplinary research approach. At the societal level, our results may have mechanistic implications for the design of institutions that aim to promote honest conduct and prevent corruption of officials with entrusted powers. Our study also provides insights for future research on corruption, such as investigating the neurocomputations required from a briber's perspective when proposing a bribe, and how the brain-synchrony between a briber and a power-holder predicts the subsequent success of bribery attempts.

## Materials and methods

### Participants

Forty Chinese-speaking undergraduates and graduate students (25 females; mean age: 20.0 ± 2.0 years, ranging from 18 to 27 years; two left-handed) were recruited via online fliers from the South China Normal University (SCNU). The sample size was determined based on previous human fMRI studies in similar topics including recent human fMRI studies on dishonesty (*Yin et al., 2017*; N = 47) and harm-aversion (*Crockett et al., 2017*; N = 28), which also adopted single-shot multiround economic decision-making paradigms. All participants had normal or corrected-to-normal vision and reported no history of psychiatric or neurological disorders. The study was performed at the Imaging Center of SCNU and was approved by the local ethics committees. All experimental protocols and procedures were conducted in accordance with the IRB guidelines for experimental testing and were in compliance with the latest revision of the Declaration of Helsinki (BMJ 1991; 302: 1194).

### Cover story

Participants were assigned the role of the power-holder who decides to accept or reject financial offers (see *Figure 1A*). They were informed that they would be presented with a series of choices

from an independent group, whose data were previously collected by the experimenter. This independent group was actually fake, and the choices made by this group were controlled by the experimenters (see Stimuli for details). Specifically, participants were led to believe that this independent group of online attendants participated in a 'Game of Chance' and had been randomly assigned to one of two roles, that is, a proposer and the third party. The proposer played either alone (i.e., the *Solo* scenario) or with a randomly matched third party (i.e., the *Dyad* scenario). In the Solo scenario, each proposer was presented with two options that would earn them different payoffs with the total of the two payoffs fixed at CNY ¥100. Similarly, in the Dyad scenario, each proposer was presented with two offers involving different combinations of payoffs divided between themself and the third party (i.e., the total amount of each payoff was also fixed to ¥100). In either scenario, one of the payoffs was randomly indicated by the computer. According to the rules of the game, the proposer should report the option indicated by the computer, which determined his/her payoff (and that of the third party if he/she was involved). However, the response of the proposer was never checked by the experimenters. This allowed the proposer to lie by reporting the alternative option that had not been indicated by the computer when this brought him/her more profits. The third party in the Dyad scenario would only be presented with the option reported by the proposer without knowing the alternative one, thus being left with no choice but to accept the monetary distribution involved in the reported option.

Importantly, the real participants were also told that each proposer had been informed that whether he or she would obtain the payoff of the reported options crucially depended on the decisions of the real participants in their role as power-holders. Therefore, to obtain the profits in the reported options, the proposer could 'share' a portion of the money from their payoff (i.e., the payoff in the reported option) to influence the power-holder. The task for the real participants was to decide whether to accept or reject the offer given the information above. If accepted, both the proposer and the participant would benefit from the payoff. However, in the Dyad scenario this could harm the interests of the innocent third party. If the participant rejected a proposition, neither the proposer nor the participant earned anything. The third party, if involved, would still be paid according to the option indicated by the computer.

## Design

We implemented a 2-by-2 within-subject design (i.e., scenario: Solo/Dyad; proposer's conduct: control/bribe) forming four experimental conditions (i.e., SC, SB, DC, and DB; see *Figure 1B*). Here, we operationally defined corrupt behaviors as the acceptance of offers proposed by the proposer in the Bribe conditions (SB and DB). In these conditions, moral costs would be elicited by conniving with the proposer to increase the payoff when the proposer lies and reports the wrong proposition, and, in the Dyad scenario, additionally harming the interests of an innocent third party. In other words, we deliberately made the potential victim (i.e., the experimenter) implicit in the Solo scenario to disentangle this kind of bribe from the one defined in the Dyad scenario where the potential victim was the third party. The other conditions (SC and DC) control the corrupt-nonspecific preference (e.g., inequity aversion) as in the typical Ultimatum game.

Each trial began with a 1.5 s information screen displaying the two payoff options together with the computer's choice (indicated by a computer icon) and the proposer's report (indicated by a blue arrow). Next, the participant was presented with an offer (proposed by the proposer) and asked to decide whether to accept or reject the offer by pressing relevant buttons with either left or right index finger within 8 s. A yellow bar appeared below the corresponding option for 0.5 s once the decision was made. If an invalid response was made (i.e., no response in 8 s or response less than 0.2 s), a warning screen showed up and this trial was repeated at the end of the scanning session. Each trial ended up with an inter-trial interval showing a jittered fixation (i.e., 3–7 s; see *Figure 1C*).

To further clarify our design, several aspects need to be noted. First, due to the potential framing effect elicited by the wording in the instruction (*Abbink and Hennig-Schmidt, 2006*), we adopted the word 'persuade' in place of words such as 'bribe' and 'corrupt' to avoid demand characteristics. Second, identities of all proposers and innocent third parties were indicated only by the initials of their names to control other confounding effects when using photos (e.g., gender, attractiveness, facial expression, dominance, and trustworthiness levels). Third, to avoid possible learning effects or strategic responses, participants were led to believe that each trial was independent and was matched with different proposers and innocent third parties. Fourth, the positions of payoff options

were randomized within participants and that of the decision options (i.e., accept or reject) were counterbalanced across participants. Fifth, participants were told that they would be paid based on the decision of only one trial (among all trials) randomly selected after the experiment. Furthermore, participants were told that the proposer (as well as the innocent third party if in the Dyad scenario) in the selected trial would also be paid accordingly.

## Stimuli

The payoffs of the proposer met the following criteria: (1) total payoffs indicated in both options (i. e., the computer-indicated option and the alternative option) were always fixed at ¥100 and (2) the payoff in the reported option by the briber was always the higher of the two for the briber (i.e., from ¥ 56 to ¥ 96 with an increment of ¥ 8). These criteria mimicked the motivation for a rational selfish proposer to lie, which increased the contextual validity of the present study. In addition, the payoffs for the innocent third party in both options (i.e., only in the Dyad scenario) were always equal to ¥100 minus the proposer's payoff in the corresponding option.

Regarding the offers (i.e., the monetary distribution between the proposer and the participant), we set the payoffs based on a fundamental principle that the proposer is selfish and always earns more for themselves in the reported options than in the alternative ones, even after bribing the power-holder (i.e., the real participant) at their own cost. The offer proportion was defined as the proportion of the amount the proposer decided to share with the power-holder from the payoff the proposer would have earned in the reported option, ranging from 10% to 90%, which in turn increases the variance of participants' responses and further benefited the computational modeling analyses. As a result, 36 different offers were adopted for the present study (see *Supplementary file 1j* for details). Each unique offer appeared once in each of the four experimental conditions (i.e., SC, SB, DC, and DB).

An event-related design was adopted and consisted of two functional runs with each containing 72 trials and lasting about 12–15 min. For each participant, we randomly divided the 36 offers into two equal datasets, then randomly associated two conditions with the first subset and two other conditions with the second subset, and assigned these stimuli to the first run (e.g., subset 1: SC, DB; subset 2: SB, DC) and the rest to the second run (e.g., subset 1: SB, DC; subset 2: SC, DB). These steps ensured that each specific bribe appears exactly twice in each run (i.e., once for two of the four conditions, respectively).

All stimuli were presented using Presentation v14 (Neurobehavioral Systems Inc, Albany, CA, USA) and were back-projected on a screen outside the scanner using a mirror system attached to the head coil.

## Procedure

Upon arrival, participants signed the written informed consent according to the Declaration of Helsinki. They were provided with the instructions for the task and completed a series of comprehension questions to ensure that they fully understood the task. Before the incentivized task, participants completed a practice session to get familiar with the paradigm and the response button. Participants were additionally informed at the beginning that the whole study included two independent tasks and the current task was always set to be the first task. To rule out the possibility of hedging the income risk across two tasks, they were informed that only one task would be randomly chosen by the computer to determine their final payoff at the end of the experiment. A 6 min structural scanning was performed at the end of the MRI session.

After that, participants filled out a battery of questionnaires. In particular, they were asked to report the subjective feeling towards (1) the moral inappropriateness of the proposer's behavior in the previous online test, (2) the seriousness of the fraud committed by the proposer, (3) the degree of their own moral conflict during decision-making, and (4) the moral inappropriateness when they accepted offers in the four experimental conditions, respectively, on a 0–100 Likert scale. In the end, participants were debriefed on all task-relevant information, informed about their final payoffs (i.e., ¥80–¥166, where ¥1 approximates to €0.13) via mobile payment, and thanked.

## Data acquisition

The imaging data were acquired on a 3-Tesla Siemens Trio MRI system (Siemens, Erlangen, Germany) with a 32-channel head coil at the Imaging Center of SCNU. Functional data were acquired using T2*-weighted echo-planar imaging sequences employing a BOLD contrast (TR = 2000 ms, TE = 30 ms; flip angle = 90°; slice thickness = 3.5 mm, slice gap = 25%, matrix = 64 × 64, FoV = 224 × 224 mm$^2$) in 32 axial slices. Slices were axially oriented along the AC-PC plane and acquired in ascending order. A high-resolution structural T1-weighted image was also collected for every participant using a 3D MRI sequence (TR = 1900 ms, TE = 2.52 ms; flip angle = 9°; slice thickness = 1 mm, matrix = 256 × 256, FoV = 256 × 256 mm$^2$).

## Data analyses

One participant was excluded due to excessive head movements (>3 mm), thus leaving a total of 39 participants whose data were further analyzed (24 females; mean age ± SD = 19.9 ± 1.9 years, ranging from 18 to 27 years; two left-handed).

## Behavioral analyses

All behavioral analyses and visualization were conducted using R (http://www.r-project.org/) and relevant packages. All reported p values are two-tailed, and p<0.05 was considered statistically significant.

For subjective rating, we implemented the within-subject MANOVA test due to the high correlation between rating measures (i.e., Pearson $r$s >0.67, ps <0.001) by the *manova* function in 'stat' package. For choice data, we performed the repeated mixed-effect logistic regression on the decision of choosing the 'accept' option by the *glmer* function in 'lme4' package, with scenario (dummy variable; reference level: Solo), proposer's conduct (dummy variable; reference level: Control), and their interaction as the fixed-effect predictors controlling for the effect of offer proportion (continuous variable; grand mean-centered) and the larger payoff the proposer would earn in the reported option (continuous variable; grand mean-centered). For the random-effect structure, we followed the 'maximal' principle (*Barr et al., 2013*) and incorporated scenario, proposer's conduct, and their interaction as the by-subject random-effect slopes. Once the regression model failed to converge, we dropped the highest interaction by-subject random-effect slope and refitted the model. In addition, we included random-effect intercepts that vary across participants and (fMRI) runs. For the statistical inference on each predictor, we performed the type II Wald chi-square test on the model fits by using the *Anova* function in 'car' package.

For the RT, we first did the log-transformation due to its non-normal distribution (i.e., Anderson–Darling normality test: A = 232.54, p<0.001) and then performed a mixed-effect linear regression on the log-transformed RT by the *lmer* function in 'lme4' package, with decision (dummy variable; reference level: accept), scenario, proposer's conduct, and their interactions (i.e., decision × scenario, decision × proposer's conduct, scenario × proposer's conduct, decision × scenario × proposer's conduct) as fixed-effect predictors controlling for the effect of offer proportion and the gain the proposer would earn in the reported option. Random-effect predictors were specified in the same way as above. We followed the procedure recommended by *Luke, 2017* to obtain the statistics of each predictor by applying the Satterthwaite approximations on the restricted maximum likelihood model fit via the 'lmerTest' package.

## Computational modeling

To provide a refined characterization of how power-holders (real participants) integrated information to determine their final decision of accepting or rejecting the offer, we tested and compared a total of seven models with different utility functions. We started from a simple model assuming that power-holders care about the offer (i.e., the monetary allocation between the proposer and oneself) differentially in terms of whether the proposer commits a bribe or not. The utility function (Model 1; *Equation 2*) is defined as follows:

$$SV(p_P, p_{\mathrm{PH}}) = \beta_P p_P + (\beta_{PH} - \theta q) p_{\mathrm{PH}} \tag{2}$$

where, in a given trial, SV denotes the subjective value, $p_P$ and $p_{\mathrm{PH}}$ represent the payoff (i.e., monetary gain) for the proposer (P) and power-holder (PH) given different choices (i.e., accept or reject

the offer), q is an indicator reflecting whether the proposer bribes (q = 1) or not (q = 0; same for models below). $\beta_P$ and $\beta_{PH}$ are two independent free parameters capturing the weights on the payoff of the proposer and the power-holder, respectively. $\theta$ describes the moral cost brought by conniving with a fraud committed by a briber (the prior range of these parameters: $-20 <= \beta_B$, $\beta_P$, $\theta <= 20$; same for models below).

Model 2 and 3 are variations established on the basis of Model 1. Specifically, Model 2 (*Equation 3*) additionally assumes that power-holders also take into consideration the unsigned inequality in the payoffs between the proposer and themselves, scaled by a free parameter $\gamma$ ($-20 <= \gamma <= 20$).

$$SV(p_P, p_{PH}) = \beta_P p_P + (\beta_{PH} - \theta q)p_{PH} + \gamma|p_P - p_{PH}| \qquad (3)$$

Model 3 (*Equation 4*) hypothesizes that power-holders bear an extra moral cost to their own payoff as a result of accepting a bribe that harms the interests of an innocent third party. This moral cost is captured by an extra parameter $\delta$ ($-20 <= \delta <= 20$). These utility functions are defined as follows:

$$SV(p_P, p_{PH}) = \begin{cases} \beta_P p_P + (\beta_{PH} - \theta q)p_{PH} & \text{if Solo scenario} \\ \beta_P p_P + (\beta_{PH} - (\theta + \delta)q)p_{PH} & \text{if Dyad scenario} \end{cases} \qquad (4)$$

Model 4 (*Equation 5*) integrates both (additional) computational components in Models 2 and 3 into the model, which is defined as

$$SV(p_P, p_{PH}) = \begin{cases} \beta_P p_P + (\beta_{PH} - \theta q)p_{PH} + \gamma|p_P - p_{PH}| & \text{if Solo scenario} \\ \beta_P p_P + (\beta_{PH} - (\theta + \delta)q)p_{PH} + \gamma|p_P - p_{PH}| & \text{if Dyad scenario} \end{cases} \qquad (5)$$

Model 5 (*Equation 1*) differs from Model 4 in the way of representing the moral cost associated with the harm to the innocent third party's interest due to the acceptance of a bribe. Thus, Model 5 assumes that power-holders take into account the exact payoff loss to the innocent third party due to the acceptance of the bribe (i.e., $p_T$ represents the payoff for the third party), which is captured by parameter $\omega$ ($-20 <= \omega <= 20$; see *Equation 1* in the Results).

We also tested the Fehr–Schmidt model (Models 6 and 7), which is adapted to the current setting. Model 6 (*Equation 6*) assumes that power-holders only consider the inequality in payoffs between themselves and the proposer, which is defined as follows:

$$SV(p_P, p_{PH}) = p_{PH} - \alpha max(p_P - p_{PH}, 0) - \beta max(p_{PH} - p_P, 0) \qquad (6)$$

where $\alpha$ and $\beta$ measure the degree of aversion to payoff inequality in disadvantageous and advantageous situations, respectively (i.e., how much power-holders dislike it when they earn less/more than the proposer). Here, we vary $\alpha$ and $\beta$ according to different conditions (i.e., $\alpha_{SC}$, $\alpha_{SB}$, $\alpha_{DC}$, $\alpha_{DB}$, $\beta_{SC}$, $\beta_{SB}$, $\beta_{DC}$, $\beta_{DB}$; $-20 <= \alpha, \beta <= 20$).

Model 7 (*Equation 7*) is established based on the three-person version of the Fehr–Schmidt model, namely assuming that participants are concerned about payoff inequality between themselves and either of the other agents in the context. The utility function is defined as follows:

$$\begin{aligned} SV(p_P, p_{PH}) &= p_{PH} - \alpha max(p_P - p_{PH}, 0) - \beta max(p_{PH} - p_P, 0) & \text{if Solo scenario} \\ SV(p_P, p_r, p_{PH}) &= p_{PH} \quad -0.5(\alpha_p max(p_p - p_{PH}, 0) + \alpha_r max(p_r - p_{PH}, 0)) \\ & \quad -0.5(\beta_p max(p_{PH} - p_P, 0) + \beta_r max(p_{PH} - p_{Pr}, 0)) & \text{if dyad scenario} \end{aligned} \qquad (7)$$

Here, we varied $\alpha$ and $\beta$ according to different agents depending on conditions (i.e., Solo scenario: $\alpha_{SC}$, $\alpha_{SB}$, $\beta_{SC}$, $\beta_{SB}$; Dyad scenario: $\alpha_{P:DC}$, $\beta_{P:DC}$ for the inequality aversion to the proposer, $\alpha_{T:DB}$, $\beta_{T:DB}$ for that to the third party).

The probability of accepting the offer was determined by the softmax function (*Equation 8*):

$$p(SV_{accept}) = \frac{e^{\tau SV_{accept}}}{e^{\tau SV_{accept}} + e^{\tau SV_{reject}}} \qquad (8)$$

where $\tau$ is the inverse softmax temperature parameter ($0 <= \tau <= 10$) denoting the sensitivity of

an individual's choice to the difference in SV between the choice of accepting the offer and that of rejecting the offer.

We fitted all the above-mentioned candidate models using the HBA approach via the 'hBayesDM' package (*Ahn et al., 2017*). The 'hBayesDM' package is developed based on the Stan language (https://mc-stan.org/), which adopts a MCMC sampling scheme to perform full Bayesian inference and obtain the actual posterior distribution. We used HBA rather than maximum likelihood estimation (MLE) method because HBA provides much more stable and accurate estimates than MLE does (*Ahn et al., 2011*). Following the approach in 'hBayesDM' package, we assumed the individual-level parameters were drawn from a group-level normal distribution: *individual-level parameters ~ Normal* ($\mu$, $\sigma$). In HBA, all group-level parameters and individual-level parameters were simultaneously estimated by the Bayes rule given the behavioral data. We fitted each candidate model with four independent MCMC chains using 1000 iterations after 2000 warm-up iterations for both studies for initial algorithm warmup per chain, resulting in 4000 valid posterior samples. Convergence of the MCMC chains was assessed through Gelman–Rubin R-hat Statistics (*Gelman and Rubin, 1992*).

For model comparison, we computed the score of leave-one-out information criterion (LOOIC) for each model (*Vehtari et al., 2017*). Compared to the point-estimate information criterion (e.g., Akaike Information Criterion), the LOOIC score offers the estimate of out-of-sample predictive accuracy in a fully Bayesian way. Conventionally, the lower LOOIC score indicates better out-of-sample prediction accuracy of the candidate model. A difference score of 10 on the information criterion scale is regarded as decisive (*Burnham and Anderson, 2004*). We selected the model with the lowest LOOIC as the winning model for subsequent analysis.

We also perform a parameter recovery analysis to ensure that our model was robustly identifiable. We first generated a simulated dataset (choices) for each participant using the individual-level posterior mean of these parameters (i.e., the true value) corresponding to that specific participant based on the winning model. Next, we fitted our winning model to the simulated dataset with the same methods (see Materials and methods for details) and obtained the individual-level posterior mean of these parameters (i.e., the recovered value). We quantified the performance of the parameter recovery by calculating the bivariate correlation, with *Pearson* correlation test, between the true value and the recovered value for each parameter, respectively.

To further examine the absolute performance of the winning model (i.e., whether the prediction of the winning model could characterize the features of real choices), we also performed a PPC (*Zhang et al., 2020*). Specifically, we generated new choice datasets, given each individual's joint posterior MCMC samples (i.e., 4000 times), in accordance with the actual trial-wise stimuli sequences presented to each participant, resulting in 4000 choices per trial per participant. Thus, for each participant, we obtained the model prediction by calculating the average offers given these new datasets in the four experimental conditions, respectively. We also performed an out-of-sample PPC to avoid the overfitting concern. Given that the stimuli (e.g., proposer's offers) in each condition were randomly but evenly distributed into two fMRI runs for each participant (see Materials and methods for details), we used the choice data from Run 1 for model estimation based on the winning model. Then, we used the mean of the posterior distribution of the individual-level parameters to simulate an independent dataset based on the stimuli used in Run 2 and calculated the acceptance rate for each participant in the different conditions (i.e., predicted acceptance rate), as we did for the real choice data in Run 2 (i.e., actual acceptance rate). For both PPC, we examined to what degree the individual difference in model prediction was correlated with that of the actual acceptance proportion using the Pearson correlation test.

## fMRI data analyses

Functional imaging data were analyzed using SPM12 (Wellcome Trust Centre for Neuroimaging, University College London, London, UK). The preprocessing procedure followed the pipeline recommended by SPM12. Functional images were first realigned to the first volume to correct motion artifacts, unwarped, and corrected for slice timing. Next, the structural $T_1$ image was segmented into white matter, gray matter, and cerebrospinal fluid with the skull removed, and co-registered to the mean functional images. Then all functional images were normalized to the MNI space, resampled with a $2 \times 2 \times 2$ mm$^3$ resolution, based on parameters generated in the previous step. The normalized images were smoothed using an 8 mm isotropic full width half maximum Gaussian

kernel. High-pass temporal filtering was performed with a default cut-off value of 128 s to remove low-frequency drifts.

## Univariate analyses

For each participant, we constructed the following GLMs to address specific aims.

Specifically, GLM1 was built to address how different types of moral cost were represented in the brain during bribery-related decision-making, which included two variants. GLM1a identified brain regions encoding the moral cost of conniving with a briber (i.e., a fraudulent proposer) on the expected gains (i.e., the profits for both the proposer and the participant had the participant accepted the offer) during bribery-related decision-making. This GLM included the onset of decision period in each condition separately as regressors of interest (i.e., SC, SB, DC, and DB). These events were modeled with the duration of the actual decision time in each trial. Furthermore, GLM1a incorporated trial-wise expected gains for both the participant and the proposer associated with each condition event as two PMs. Notably, the default orthogonalization process on the PMs was switched off, allowing that these two PMs competed for variance during the estimation. We also performed a supplementary GLM analysis (GLM1a-s) to explore regions that encode the above PMs across all trials. GLM1b aimed at investigating regions tracking the losses to the innocent third party because of corruption. It was constructed similarly to GLM1a, except that only the onset of the decision period in DB condition was associated with the trial-wise expected loss to the third party (i.e., the absolute payoff difference of the innocent third party had the participant accepted the offer) as the sole PM. We did not add the same PM to the DC condition as the payoff to the innocent third party would not change as a result of the participant's decision (i.e., the third party would receive the payoff indicated in the computer-chosen option anyway) and the expected loss would always be 0. Notably, we chose not to incorporate the PM of expected loss to the third party into the GLM1a because this GLM aims to identify brain regions specifically encoding the expected gains due to bribe-taking. For this reason, it is better to keep the design matrix balanced so that each onset regressor of the decision event is attached with the exact same PMs (i.e., PM of expected gains for the proposer and the participant). To control for the possible effect of these PMs on the result, we implemented a supplementary GLM analysis (i.e., GLM1b-s) in which we not only added the PM of the expected loss (the first PM) but also incorporated the PMs of expected gains for the proposer and the participant to the DB condition (the second and third PM). The orthogonalization was performed to control for the co-linearity between these PMs (see *Supplementary file 1k*). We also established GLM1c, which did not contain any PM for subsequent functional connectivity and multivariate analyses.

GLM2 identified regions computing relative SV (i.e., relative SV) during bribery-related decision-making. The relative SV was defined by subtracting the SV of the non-chosen option from that of the chosen option (i.e., relative SV = $SV_{chosen}$ - $SV_{unchosen}$) based on the winning model. GLM2 consisted of two variants. GLM2a distinguished the Bribe condition from the Control condition (pooling the Solo and Dyad scenarios), which contained the onset of relevant trials during the decision period with the trial-wise relative SV as the associated PMs as the regressor of interest. GLM2b was similar to GLM2a except that it pooled all trials as the single regressor of interest, which examined a general value-related neural network regardless of experimental conditions.

GLM3 examined the neural activities during bribery-related decision-making with regard to specific choices. Thus, GLM3 contained the onset of decision period of each condition with respect to specific choices (i.e., *accept* or *reject* in conditions of SC, SB, DC, DB), resulting in eight regressors of interest. To control the differential effect of SV on trials with regard to specific choice, we also attached each onset regressor with corresponding trial-wise relative SV as PMs. Notably, six participants had to be further excluded from this specific GLM analyses due to the missing type of choices in one or more of the four conditions. To simplify further analyses, we computed the neural activity specific to rejecting as opposed to accepting offers (i.e., reject vs. accept) in each of the four conditions.

For all GLMs above, we also separately modeled the onsets of information period for each condition, together with the onset of the button press. Furthermore, once the participant showed invalid responses, a regressor modeling events of no interest was included, which contained decision onsets of invalid trials (i.e., for trials for which RT of making choices was less than 200 ms, duration equals

the actual RT; for trials of no response, duration equals 8 s) as well as the warning feedback (duration equals 1 s). Onsets of these events were regarded as regressors of no interest. The six movement parameters were added to all models as covariates to account for the artifacts of head motion. The canonical hemodynamic response function (HRF) was applied to model the fMRI signal.

Individual-level contrasts were fed to group-level random-effect analyses. One-sample $t$-tests were mainly adopted to test the parametric effect or to compare simple contrasts between two conditions. A $2 \times 2$ within-subject flexible factorial ANOVA model was also adopted to examine the main effect of scenario (i.e., Dyad vs. Solo), the main effect of the proposer's conduct (i.e., Bribe vs. Control), and their two-way interaction (i.e., (DB – DC) – (SB - SC)) on neural activity while applicable (see Results for details of the tests we used for specific analyses).

For all univariate analyses, we adopted a whole-brain corrected threshold of $p<0.05$ at the cluster-level controlling for family-wise error (FWE) rate with an uncorrected voxel-level threshold of $p<0.001$ as the cluster-defining threshold (*Eklund et al., 2016*). Moreover, a small volume correction (SVC) was conducted within these hypothetical regions of interest (ROI). All masks were defined based on a whole-brain parcellation given a meta-analytic functional coactivation map of the Neurosynth database (https://neurovault.org/collections/2099/). Compared with an ROI definition based on specific peak coordinates, this approach generates masks that are much less biased by a specific scientific research (e.g., an empirical study or a meta-analysis) and are larger in volume (vs. a sphere with a radius of 6–8 mm). This ensures the validity and the reliability of the SVC. Regions were labeled according to the AAL template via the xjView toolbox (http://www.alivelearn.net/xjview8/). To illustrate the parametric effect of potential loss for the third party (GLM1b, GLM1b-s), and relative SV (GLM2a), we adopted the rfxplot toolbox (*Gläscher, 2009*) (http://rfxplot.sourceforge.net/).

## Multivariate decoding analyses

To investigate whether TPJ/pSTS was selectively involved in bribery-related decision-making that harmed a third party, we performed an ROI-based multivariate decoding analysis via python-based nltools package (v 0.3.6 in Python 3.7.1; http://github.com/ljchang/nltools). Specifically, we trained a linear support vector machine (slack parameter C = 1 chosen as default) classifier, a widely used algorithm to deal with the binary classification, to discriminate decision-relevant activity pattern in TPJ/pSTS in Bribe vs. Control for the Solo (i.e., individual contrast maps in GLM1: SB vs. SC; the first condition was coded as 1 and the second as −1, same below) and the Dyad scenario (DB vs. DC) separately.

A LOSO cross-validation procedure was adopted such that it trained the classifier on N-1 participants and generated a weight map that best classified the sample, and then tested the classification on the left-out participant. This procedure looped for all participants once to obtain their respective cross-validated signature response values. As a result, the classifier obtained a hyperplane that best discriminates the individual contrast maps in two conditions (e.g., SB vs. SC). Receiver operating characteristic curve was created based on the performance of classification (i.e., the two-choice forced-alternative accuracy) via Canlab Matlab toolbox (https://github.com/canlab). Statistical significance was obtained via the permutation test (i.e., 5000 times of permutation).

## Pattern similarity analyses

To further examine whether vmPFC was engaged in value computation during decision-making in both the Bribe and the Control conditions, we implemented a multivariate pattern similarity analysis with the nltools package.

For each participant, we extracted the multivoxel patterns within the vmPFC mask (defined based on the Neurosynth coactivation map; see above for details) from parametric contrasts of relative SV in the Bribe and the Control conditions (GLM2a). Then we calculated the similarity between these two patterns via the Pearson correlation and performed Fisher's z transformation on these correlation coefficients for statistical analyses. Group-level statistical significance was obtained via the permutation-based one-sample $t$-test (i.e., 5000 times of permutation).

## Functional connectivity analyses

To explore how the value signal in the vmPFC (i.e., the relative SV effect in GLM2a) interacts with other parts of the brain during the bribery-related decision-making process, we implemented a PPI

analysis (*Friston et al., 1997*) using the gPPI toolbox (https://www.nitrc.org/projects/gppi) (*McLaren et al., 2012*). To this end, for each participant, we constructed a PPI-GLM containing the following regressors: (1) the de-convolved time series in the vmPFC within the parcellation-based vmPFC mask as the physiological regressor, (2) all onsets of the decision period in GLM1c as the psychological regressors, and (3) the generated PPI regressors by multiplying the physiological regressor with each psychological regressor. These regressors were all convolved with the canonical HRF to model the BOLD signal. In addition, we also incorporated six movement parameters as covariates to account for artifacts of head motion. Given the purpose of this analysis, we focused on the PPI contrast of Bribe vs. Control with dlPFC as our target region and examined whether the dlPFC enhanced functional connectivity with vmPFC during the bribery-related decision-making.

## Inter-subject representational similarity analysis (IS-RSA)

The IS-RSA was treated as an exploratory analysis with a focus on dlPFC and conducted using the nltools package. We first computed the corruption-relevant contrast (Bribe vs. Control) during the decision period derived from GLM1c. Next, we extracted the parameter estimates (i.e., contrast values) of all voxels from the bilateral dlPFC (i.e., neural activity patterns; see above for details of mask selection) and constructed neural representational dissimilarity matrices (neural RDM) using pairwise correlation dissimilarity of these neural patterns between each pair of participants. To characterize bribery-specific preferences across participants, we also created a parameter RDM that measures the Euclidean distance between each pair of participants in a parameter space reflecting two forms of bribery-specific moral costs (i.e., $\theta$ and $\omega$). Then, we computed the correlation between the parameter RDM and the neural RDMs using Spearman's rank-order correlation, which did not assume a linear behavior–brain relationship. Statistical significance was obtained via the permutation *t*-test (i.e., 5000 times of permutation).

## Acknowledgements

This research has benefited from the financial support of IDEXLYON from Université de Lyon (project INDEPTH) within the Programme Investissements d'Avenir (ANR-16-IDEX-0005) and of the LABEX CORTEX (ANR-11-LABX-0042) of Université de Lyon, within the program Investissements d'Avenir (ANR-11-IDEX-007) operated by the French National Research Agency. This work was also supported by grants from the Agence Nationale pour la Recherche and NSF in the CRCNS program to JCD (ANR n°16-NEUC-0003-01), National Science Foundation of China to QC (31470995), and China Postdoctoral Science Foundation to YH (2019M660007). We thank Sima Hakimi, Zixuan Tang, Siying Li, and Yaner Su for helpful assistance with data collection. We also thank Dr. Xiaoxue Gao for the assistance of implementing the multivariate fMRI analyses and Zhewen (Vane) He for proofreading the early draft of the manuscript.

## Additional information

### Funding

| Funder | Grant reference number | Author |
| --- | --- | --- |
| Agence Nationale de la Recherche | ANR-16-IDEX-0005 | Jean-Claude Dreher |
| China Postdoctoral Science Foundation | 2019M660007 | Yang Hu |
| National Natural Science Foundation of China | 31970982 | Chen Qu |
| Agence Nationale de la Recherche | ANR-11-LABX-0042 | Jean-Claude Dreher |
| Agence Nationale de la Recherche | ANR-11-IDEX-007 | Jean-Claude Dreher |
| Agence Nationale de la Recherche | ANR n°16-NEUC-0003-01 | Jean-Claude Dreher |

The funders had no role in study design, data collection and interpretation, or the decision to submit the work for publication.

### Author contributions
Yang Hu, Conceptualization, Resources, Data curation, Software, Formal analysis, Investigation, Visualization, Methodology, Writing - original draft, Writing - review and editing; Chen Hu, Formal analysis, Methodology, Writing - review and editing; Edmund Derrington, Brice Corgnet, Writing - review and editing; Chen Qu, Conceptualization, Data curation, Supervision, Funding acquisition, Validation, Writing - original draft, Project administration, Writing - review and editing; Jean-Claude Dreher, Conceptualization, Data curation, Supervision, Funding acquisition, Validation, Methodology, Writing - original draft, Project administration, Writing - review and editing

### Author ORCIDs
Yang Hu (iD) https://orcid.org/0000-0001-7659-5782
Chen Hu (iD) http://orcid.org/0000-0003-2289-743X
Chen Qu (iD) https://orcid.org/0000-0002-8465-8007
Jean-Claude Dreher (iD) https://orcid.org/0000-0002-2157-1529

### Ethics
Human subjects: The study was performed at the Imaging Center of SCNU and was approved by the local ethics committees. All experimental protocols and procedures were conducted in accordance with the IRB guidelines for experimental testing and were in compliance with the latest revision of the Declaration of Helsinki (BMJ 1991; 302: 1194). Upon arrival, participants signed the written informed consent.

### Decision letter and Author response
Decision letter https://doi.org/10.7554/eLife.63922.sa1
Author response https://doi.org/10.7554/eLife.63922.sa2

## Additional files
### Supplementary files
- Supplementary file 1. Supplementary tables.
- Transparent reporting form

### Data availability
All relevant data and codes have already been uploaded to an open data depository (https://github.com/huyangSISU/2021_Corruption, copy archived at https://archive.softwareheritage.org/swh:1:rev:97a721939333b7da7123baa6258057bdddfec5ef/).

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
