## [Decision Letter]

**Acceptance summary:**

This manuscript will be of interest for social psychologists and neuroscientists as it presents a first step toward understanding the neural correlates of decisions that involve corruption and harm third parties. The behavioral and neuroimaging data support the conclusion that different moral costs associated with such choices are correlated with activity in different brain areas (anterior insula and temporo-parietal junction), but integrated into a value signal in the ventromedial prefrontal cortex.

**Decision letter after peer review:**

Thank you for submitting your article "Neural basis of corruption in power-holders" for consideration by *eLife*. Your article has been reviewed by three peer reviewers, and the evaluation has been overseen by a Reviewing Editor and Michael Frank as the Senior Editor. The reviewers have opted to remain anonymous.

The reviewers have discussed the reviews with one another and the Reviewing Editor has drafted this decision to help you prepare a revised submission.

Summary:

This human fMRI study focuses on the behavioral and neural correlates of decisions involving fraud and corruption. In a novel behavioral task, subjects could either accept or reject offers from a proposer who divided a monetary payoff in a recommended or fraudulent way. The latter involves a bribe to the participant as well as harm to a third party. The key finding is that the two types of moral costs at stake here (norm violation and harming a third-party) correlate with activation in different brain regions (insula and TPJ).

All reviewers agreed that this study addresses an important question and that it is well executed, with an impressive set of analytical tools. They also agreed that the manuscript will be of interest to a broad readership. However, there was consensus on several important issues that need to be addressed in a revised version of the manuscript.

Revisions:

Reviewers identified several essential issues that need to be addressed before the manuscript can be accepted for publication in *eLife*:

1) There is a concern about selective reporting and cherry-picking. Findings should be reported more comprehensively in the main text and SI. And clarification is needed regarding how correction for multiple comparisons was applied (ROIs, models, etc.).

2) There were concerns about the lack of random effects in the mixed linear models for the analysis of behavioral data.

3) It would be important to present data to confirm that subjects believed the cover story as a basic manipulation check of the experiment.

4) Please include a more detailed plan for data and code sharing.

5) Please revise the framing of the manuscript.

6) The individual differences analysis is likely underpowered and the results should clearly be described as exploratory.

Reviewer #1:

This manuscript assesses the behavioral and neural correlates of decisions to accept an offer from a proposer who either follows a monetary payoff recommended by the computer (control condition) or lies by choosing the option not recommended by the computer (bribe condition). In addition, in a solo scenario, the proposer's offer to the participant does not harm anyone, while in a dyad condition it harms a third party whose payoff will be decreased as a consequence of the proposer behaving dishonestly by not following the computer choice. The authors report that these two moral costs (the proposer "lying" and the third party being harmed) both reduce the participants' propensity to accept the proposer's offer, and can be captured by two parameters in a computational model. Finally, a combination of univariate and multivariate analyses of the neuroimaging data is reported to identify how some of the model-predicted signals are encoded in the brain, and particularly in the anterior insula, TPJ, vlPFC and dlPFC.

This study is impressively executed, the manuscript is clearly written, and the topic of moral transgression and integration between dishonest behavior and third-party harm is novel and very relevant. However, I still have concerns that I would like to see addressed before I can recommend this manuscript for publication.

1) Study framing and ecological validity

a) Given the current climate in the world of misinformation spreading and the media tendency to misinterpret scientific results and jump to conclusions, I would recommend the authors to use a title and a framing that reflects more precisely the findings of their study. As they acknowledge, corruption is a very complex process, and what their task assesses is a small part of what can lead to corruption, namely the role of two forms of moral transgressions (fraud and third-party harm) in the decision to accept a bribe, and their representation and integration in the brain. Generally, I would refrain from using such a strong term as corruption, except maybe in the Discussion where the implications of the findings in light of understanding corruption can be brought up.

b) This concern mostly stems from the overall lack of ecological validity of the task used. Specifically, the proposer's behavior was fully controlled by the experimenter (and a cover story was used to pretend otherwise) and it is unclear whether participants fully believed the cover story or not. Was participants' belief that proposers and third parties were attendants of a previous study actually tested? If participants don't believe this cover story and instead suspect that everything is fictitious, their behavior would not constitute a moral transgression. In the dyad situation, it also seems like the third party will never know that they actually would have gotten a better outcome if the proposer hadn't lied. If that's indeed the case, that would make the current task less likely to mimic real-life situations where third parties are aware they were harmed (e.g. competitors for a project who do not get selected).

c) If I understand the design correctly, there was never a situation where the computer picked the low payoff option and the proposer also honestly reported this offer. This appears to me as a major drawback of the task as the participant could interpret the proposer's behavior as being simply value maximizing rather than fraudulent or dishonest. Also, while helpful for the analysis, the fact that essentially every single trial contained a bribe is concerning. If proposers' choices had been obtained in a previous online study, they would have likely looked very different than what is being displayed to the participants in the current task. Presumably, many proposers would have followed the computer choice even for the low payoff option, and/or would not have offered to split their offer by the indicated proportions. Additionally, it is likely that the share offered by the proposer would be higher in the bribe than in the control condition. I believe all these concerns should be acknowledged in the Discussion.

2) fMRI analysis and interpretation of results

Generally, I find that the fMRI results lack cohesion and a clear interpretation that tie them together, mostly due to the combination of many methods/GLMs used and not always justified, to the unclear process of selecting and using multiple regions of interest, and to potential confounds in the contrasts and regressors examined.

a) How many regions of interests were included? The authors state in the Materials and methods that they used a whole-brain cluster-level FWE correction at P<0.05, but most the key results are in fact small-volume corrected. It seems very unlikely that the authors had only one a priori region of interest in mind for each analysis they report. In the Introduction for example, they seem to focus on four key regions (vAI, TPJ, vmPFC, dlPFC), but other studies on dishonesty have reported potential roles of the amygdala (Garrett et al., 2016, Nature Neuroscience), or nucleus accumbens (Speer et al., 2020, PNAS) in dishonest behavior. Were other ROIs considered initially? If so, correction for the number of ROIs may be needed.

b) Given the design of the task, any activation contrasting the bribe vs control condition cannot distinguish between representing the proposer lying vs being honest and the computer recommending the low vs high payoff offer, as the proposer always lies when the computer chooses the low payoff. Because of this, I find the results in Figure 3A difficult to interpret. Why would we expect vAI to track expected personal profits positively in the DB condition but negatively in the SC condition, and not at all in the other two conditions? There also seems to be a main effect of scenario (dyad vs solo) on vAI tracking on expected profits, which is not discussed.

c) Why wasn't the expected loss to the third party added as a parametric modulator of the DB condition in GLM1a, thus allowing to control for the expected gains of the participant and the proposer? Is this because the loss to the third party is highly correlated with the gain of the proposer? If so, then those two signals can't be separated, and this should be addressed.

d) The rationale for examining vmPFC-dlPFC functional connectivity analyses is unclear to me. Was connectivity with other regions of interest, like TPJ or vAI, tested but not significant? If so, the authors should be clear about this and correct for the number of regions tested. Similarly, the Materials and methods section about the inter-subjects RSA suggests that several "hypothesized regions including bilateral dlPFC" were tested, but then the results focus exclusively on the dlPFC. If other regions were indeed tested, this should be clarified and accounted for.

e) The interpretation of the IS-RSA results in the Discussion is unclear, especially in what those results mean in and of themselves, and how they can be reconciled with the univariate dlPFC result and the vmPFC-dlPFC functional connectivity analysis.

Reviewer #2:

The reported study investigates the neural basis of corruption in social interactions. The main finding is that two types of moral costs in bribery (norm violation versus harming a third-party) correlate with activation in dissociable brain regions (insula versus TPJ). There is much to like about this well-written manuscript: it addresses an important question, and the experimental manipulations appear sound. The manuscript will certainly be of interest to a broad readership working on social interactions or on the neural basis of decision making. Nevertheless, I have a couple of concerns, particularly regarding data analysis, which the authors need to address in a revision.

1) The GLMs used for the fMRI analyses should be specified in more detail already in the Results section, not only in the Materials and methods. This would make it easier to understand which contrasts show significant activation in insula or TPJ. In general, I find the presentation of the imaging results rather confusing, mainly because a large number of analyses was computed (nine in total: GLM1a-c, GLM2a-b, GLM3, PPI, multivariate analysis, IS-RSA), and the authors often just selectively report one contrast of each GLM, while the results of other contrasts are not even reported in supplementary tables. This analysis approach raises questions regarding the robustness of the imaging results. For example, when testing the hypothesis that the TPJ shows enhanced activation in dyad versus solo scenarios, no significant results are observed in two contrasts for GLM1c, but only in an additional multivariate analysis. However, these non-significant results are ignored in the Discussion section, and the significant TPJ finding in the multivariate analysis is taken as evidence for the authors' hypothesis. It also remains unclear why the authors specified separate models with parametric modulators for personal benefits (GLM1a) and third-party loss (GLM1b), as in principle these variables could be modelled within one model. All this leaves the impression of cherry-picking and makes the imaging results appear less robust and convincing than as they are presented.

2) Regarding the mixed linear models for the analysis of behavioral data, the authors state that "factors allowing varying intercept across participants" were entered as random-effect predictors. Please be more precise regarding which random effects were specified in the model. It is generally recommended to maximize the random effects structure in order to minimize the risk of type I errors (Barr et al., 2013). As the current study seems to follow a strict within-subject design, I think that all fixed-effect predictors should be modelled as random slopes in addition to random intercepts

3) The task did not involve real social interactions, but the offers were computer-generated. I might have missed it, but it seems nowhere stated whether participants believed the cover story or not (was this assessed at all). In any case, the authors should add a caveat in the Discussion section clarifying that the social interactions in the study were only hypothetical.

Reviewer #3:

Dr. Hu et al. report a neuroimaging study of corruption in which (computer agent) proposers provide participants the opportunity to personally benefit from turning a blind eye to deception that in some cases has monetary costs for a third party. Through a series of computational models, the authors confirm that participants incur a moral cost, beyond inequity models, for engaging in the corrupt act, that is at its worst when a third party is injured. Each component of the model is then tied to a specific aspect of brain function that aligns with prior findings. The authors conclude that an inhibition mechanism is associated with reduced participation in corrupt acts.

I enjoyed the paper. It covers an interesting topic, is methodical, and shows a great deal of expertise in a wide range of methods.

1) Tests for the involvement of a particular brain region in a given step of the corruption decision are mixed without justification. GLM and MVPA analyses seem to be deployed as tests with increasing sensitivity rather than to test for computational differences. Not making a distinction is understandable since there is not a consensus on how each should be interpreted. However, this sort of variation on testing until a finding is reached can lead to overestimation of effect sizes and confirmation bias of previously proposed roles. Is there a reason beyond sensitivity for using multivariate models in some cases rather than others?

2) A number of GLMs are used to reach different conclusions presumably because of covariance issues between regressors. This, in and of itself, is not a problem if the covariance tables are shown and the repartitioning of common variance is acknowledged/interpreted.

3) Individual difference model – individual difference studies of forty participants assume a large effect size to be considered reliable. Most psychological (and biological indices) do not fall in this range. The result is fine as an exploratory analysis but that means it should be described as such.

4) Deception success measures. Was any data collected to confirm the believability of the proposer and third-party deception? Were participants debriefed afterwards?

5) The brain data is largely used to lend construct validity to the corruption task and confirm psychological interpretations of the processes involved in the decision. Ideally, the neural data would be used to adjudicate between two competing behavioral models (and/or behavioral data used to adjudicate between competing neural models). This somewhat lessens the utility of a promising neuroimaging dataset.

6) Complete analysis scripts (Main and SI, behavioral and neuroimaging) and minimally processed imaging data should be posted and referenced for the article. It is not clear that these fit in the 'source data' option for *eLife* without an accession number noted in the article.

---

## [Author Response]

Revisions:Reviewers identified several essential issues that need to be addressed before the manuscript can be accepted for publication in eLife:1) There is a concern about selective reporting and cherry-picking. Findings should be reported more comprehensively in the main text and SI. And clarification is needed regarding how correction for multiple comparisons was applied (ROIs, models, etc.).2) There were concerns about the lack of random effects in the mixed linear models for the analysis of behavioral data.3) It would be important to present data to confirm that subjects believed the cover story as a basic manipulation check of the experiment.4) Please include a more detailed plan for data and code sharing.5) Please revise the framing of the manuscript.6) The individual differences analysis is likely underpowered and the results should clearly be described as exploratory.

We thank the editor for summarizing these points which are key to our revisions. First, regarding the comprehensive report of our results, we have included new tables to describe them (see Supplementary File 1F-1I) and we have added more details to clarify how we implemented the ROI-based analyses (point #1). We have also performed additional analyses to address the random-slope issue (point #2). We agree with the reviewer’s concern about analyses of individual differences, and we have therefore re-framed this part as exploratory analyses (point #6). Regarding issues relevant to the cover-story and manuscript framing, we do admit the importance of these issues and we have incorporated them into the revised manuscript for an extensive discussion. We have also made a detailed response to justify the validity of the experimental setup (point #3) and the reason why we think the current framing worked (point #5). Last but not least, we fully embrace the open-and-transparency policy advocated by *eLife* and have already uploaded the most relevant data and codes to an open data depository copy archived at https://github.com/huyangSISU/2021_Corruption which will be public for readers who are interested in this study once the paper is accepted (point #4).

Reviewer #1:This manuscript assesses the behavioral and neural correlates of decisions to accept an offer from a proposer who either follows a monetary payoff recommended by the computer (control condition) or lies by choosing the option not recommended by the computer (bribe condition). In addition, in a solo scenario, the proposer's offer to the participant does not harm anyone, while in a dyad condition it harms a third party whose payoff will be decreased as a consequence of the proposer behaving dishonestly by not following the computer choice. The authors report that these two moral costs (the proposer "lying" and the third party being harmed) both reduce the participants' propensity to accept the proposer's offer, and can be captured by two parameters in a computational model. Finally, a combination of univariate and multivariate analyses of the neuroimaging data is reported to identify how some of the model-predicted signals are encoded in the brain, and particularly in the anterior insula, TPJ, vlPFC and dlPFC.This study is impressively executed, the manuscript is clearly written, and the topic of moral transgression and integration between dishonest behavior and third-party harm is novel and very relevant. However, I still have concerns that I would like to see addressed before I can recommend this manuscript for publication.1) Study framing and ecological validitya) Given the current climate in the world of misinformation spreading and the media tendency to misinterpret scientific results and jump to conclusions, I would recommend the authors to use a title and a framing that reflects more precisely the findings of their study. As they acknowledge, corruption is a very complex process, and what their task assesses is a small part of what can lead to corruption, namely the role of two forms of moral transgressions (fraud and third-party harm) in the decision to accept a bribe, and their representation and integration in the brain. Generally, I would refrain from using such a strong term as corruption, except maybe in the Discussion where the implications of the findings in light of understanding corruption can be brought up.

This is a critical point. On the one hand, we fully agree with the reviewer that the present study only adopted a simplified setting in the lab to simulate a certain type of corruption, which is actually a complicated issue that involves various forms and complex cognitive processes. We are also aware of the reviewer’s concern about possible misinterpretation by the media and its potential consequences. On the other hand, we believe that our design has successfully captured the essence of corruption, especially from the perspective of a person in power, and transferred it to an experimentally testable situation. As we stated in the Introduction, our design introduced an interpersonal context where a power-holder and a briber form a reciprocal relationship so that they can earn morally-tainted benefits together, sometimes at the expense of a third party. All these points are key components in defining bribe-taking behavior and distinguishing it from other types of immoral behaviors such as dishonesty, betrayal or aggression. Moreover, the inclusion of the Control condition reinforces that the observed behavioral and neural effects were specific to bribery-related decision-making rather than social decision-making in general.

Furthermore, given that almost all neuroimaging studies investigating the neural basis of a given cognitive process in social contexts are performed in a lab environment, changing the current title to something like ‘Neural basis of corruption in a lab setting’ is a statement of the obvious and would actually restrict our topics. In addition, if we focused on only the moral costs brought by bribe-taking and modify the title to something like “Neural basis of fraud and harm incurred by a third party during bribery-related decision-making”, it becomes needlessly too specific.

Taken together, we still believe that the framing of the current manuscript fits the theme and the scope of corruption. However, we do respect the reviewer’s concern, and we have now incorporated these points into the Discussion as a limitation, which reads as follows:

“Several issues concerning the present study need further discussion. First, although the present task captures the essence of corruption from the perspective of a person in power, it only simulates a specific type of corruption in a lab setting. The real phenomenon is far more complex and diverse. This obviously constrains the generalization of the current findings to corruption in field settings …”

Additionally, we will be very careful when communicating to the media not to misinterpret our results and to restrict its application to a lab situation.

b) This concern mostly stems from the overall lack of ecological validity of the task used. Specifically, the proposer's behavior was fully controlled by the experimenter (and a cover story was used to pretend otherwise) and it is unclear whether participants fully believed the cover story or not. Was participants' belief that proposers and third parties were attendants of a previous study actually tested? If participants don't believe this cover story and instead suspect that everything is fictitious, their behavior would not constitute a moral transgression. In the dyad situation, it also seems like the third party will never know that they actually would have gotten a better outcome if the proposer hadn't lied. If that's indeed the case, that would make the current task less likely to mimic real-life situations where third parties are aware they were harmed (e.g. competitors for a project who do not get selected).

The present task adopts a context without involving real interpersonal interactions (regardless of real partners or counterparts). Instead, we took a multi-round single-shot economic game, which is commonly used in neuroeconomics studies (e.g., Spitzer et al., 2007; Aimone et al., 2014; Hu et al., 2018). Compared with the interactive games that involve the same partner (or a few partners), repeatedly occurring during the experiment, the multi-round single shot games usually introduce a scenario in which participants are presented with choices collected from another independent group in a previous experiment. Although this kind of task, to some extent, reduced the ecological validity as the reviewer pointed out, we chose to use it here because each choice can be considered independent and this reduces the confounding effects of learning or concerns of reputation via repeated interactions with the same partner that might lead to different computations and neural mechanisms (e.g., strategic decision-making).

Given the experiment setting (or cover story) of the multi-round one-shot task, it would be impossible to invite all “partners” to the lab (i.e., proposers and third parties in the current task) to prove their existence, no matter whether they are real partners, confederates, or fictitious agents. In our view, these problems make such a belief test unfeasible. Moreover, asking such a question at the end of the experiment might also cause other unexpected issues. For example, participants who actually believe the cover story during the experiment might be prompted to doubt the cover story, simply because they were asked about it. For these reasons, studies in the literature of neuroeconomics that use this kind of task do not usually elicit participants’ belief after the experiment (see references mentioned above).

Following this tradition of previous studies, we did not explicitly ask whether participants believed our cover story. However, several lines of evidence converge to indicate that they actually believed it: (1) Before the task, all participants successfully passed the comprehension quiz; (2) The observed behavioral pattern (i.e., choices in the fMRI task and subjective rating afterward) conformed to our prediction that was built based on the assumption that participants took into account the morality of the proposer’s (briber’s) offer and the harm to the third party; (3) None of the participants raised any doubts with respect to the reality of the cover story when we debriefed them at the end of the experiment. In spite of that, we do think the lack of real social interactions in the present study is an important point to address and thus we have added it to the Discussion as follows:

“…Third, the present task adopted a multi-round single-shot economic game that did not involve any real partner physically present during the experiment. We decided to use such an experimental setting not only because it is commonly used in neuroeconomics studies (e.g., Spitzer et al., 2007; Aimone et al., 2014; Hu et al., 2018) but also because, here, each choice can be considered independent, thus alleviating the potentially confounding effects of learning and concerns of reputation in repeated interactions. Although the lack of social interactions in the task might diminish the involvement of participants and influence their beliefs about the authenticity of the experimental setting, no participants explicitly raised doubts about the reality of the cover story and our behavioral results were consistent with the predictions which were built on the assumption that participants believed the existence of proposers and third parties. Nevertheless, the concerns raised above cannot be completely ruled out …”

The Dyad scenario was designed in terms of the sender-receiver game, a classical economic game used for investigating deception (for details of the game, see Gneezy 2005). In this game, only the sender (i.e., the proposer in the current task) was informed about the full information of the payoffs, which was unknown to the receiver (i.e., the third party in the current task). While in some cases, as the reviewer pointed out, third parties are aware they have been harmed because of corruption, it is also very frequent that third parties remain ignorant of bribery-related harm. This is especially the case when third parties have lower social status, which isolates them from the truth behind the scenes. For instance, a real estate developer could bribe a government official who may turn a blind eye when the developer uses substandard materials on public housing development. The families who subsequently live in these apartments would remain ignorant that they had been cheated and those substandard materials might harm their quality of life and even health.

c) If I understand the design correctly, there was never a situation where the computer picked the low payoff option and the proposer also honestly reported this offer. This appears to me as a major drawback of the task as the participant could interpret the proposer's behavior as being simply value maximizing rather than fraudulent or dishonest. Also, while helpful for the analysis, the fact that essentially every single trial contained a bribe is concerning. If proposers' choices had been obtained in a previous online study, they would have likely looked very different than what is being displayed to the participants in the current task. Presumably, many proposers would have followed the computer choice even for the low payoff option, and/or would not have offered to split their offer by the indicated proportions. Additionally, it is likely that the share offered by the proposer would be higher in the bribe than in the control condition. I believe all these concerns should be acknowledged in the Discussion.

As the reviewer pointed out, the current task did not contain a condition where the proposer honestly reported the option with a lower payoff randomly selected by the computer. While it would definitely be possible to add this kind of trial, we did not do this purposely to maintain the symmetry of the experimental setting. For the sake of a balanced design, such trials should be incorporated into both the control and bribe condition. However, this would result in completely irrational behavior by the proposer if we were to do so. Take the Solo scenario for example. While the proposer’s motivation is clear in the Control condition when honestly reporting a computer-chosen disadvantageous option (i.e., the proposer honestly chose less profits for himself), this would lead to a counter-intuitive situation in the Bribe condition when the proposer chose a disadvantageous option when an advantageous option was selected by the computer (i.e., the proposer cheated to obtain less profits for himself). Such situations would make the cover story bizarre because it would be very difficult for participants, as power-holders, to understand the motivation underlying such “white lies” in this situation (i.e., cheating to benefit others). Furthermore, adding such trials in either condition to the current task would make it meaningless to propose offers (bribes) for the power-holder because doing so would make the proposer earn even less (compared with reporting the option with a larger payoff for himself). Thus, the inclusion of such trials would have conflicted with the logic of a (fictive) proposer who always wants to optimize his gains. In addition, adding these trials would inevitably prolong the duration of the experiment, fatiguing participants more, and thus perhaps reducing the quality of the data we collected.

Despite these reasons for not including such trials, we agree with the reviewer’s concern about the potential drawback of the present design, and we therefore added these points to the Discussion, which now reads as follows:

“… Second, the present task did not contain a condition in which the proposer honestly reported the option with the lower payoff having been randomly selected by the computer. We did this purposely to preserve the symmetry of the experimental design and to maintain the motivation of proposers’ behavior to being entirely rational and explainable by the optimization of personal profits. Thus the offer proposition, in either the Bribe or the Control condition, was realistic for the proposer, i.e., it would not make sense if the proposer proposes an offer to earn a lower payoff. In addition, adding such trials to the current task would inevitably prolong the duration of the experiment, which might make participants more tired and thus influence the quality of the data we collected.”

2) fMRI analysis and interpretation of resultsGenerally, I find that the fMRI results lack cohesion and a clear interpretation that tie them together, mostly due to the combination of many methods/GLMs used and not always justified, to the unclear process of selecting and using multiple regions of interest, and to potential confounds in the contrasts and regressors examined.a) How many regions of interests were included? The authors state in the Materials and methods that they used a whole-brain cluster-level FWE correction at P<0.05, but most the key results are in fact small-volume corrected. It seems very unlikely that the authors had only one a priori region of interest in mind for each analysis they report. In the Introduction for example, they seem to focus on four key regions (vAI, TPJ, vmPFC, dlPFC), but other studies on dishonesty have reported potential roles of the amygdala (Garrett et al., 2016, Nature Neuroscience), or nucleus accumbens (Speer et al., 2020, PNAS) in dishonest behavior. Were other ROIs considered initially? If so, correction for the number of ROIs may be needed.

As reported in the Introduction, we adopted a total of 4 ROIs (vAI, TPJ, vmPFC and dlPFC) based on clear hypotheses with regard to the specific analyses throughout the whole study. Please allow us to re-explain our key hypotheses. First, we investigated how different forms of moral costs brought by bribe-taking were encoded in the brain of a power-holder. Specifically, we came up with two analyses. On the one hand, we tested whether the moral cost of conniving with a fraudulent proposer altered the valuation of the expected personal gains due to the acceptance of the bribe (GLM1a). In the moral domain, vAI is known to be engaged when social norms or moral principles are violated. For instance, a stronger vAI signal has been observed when people are treated unfairly (Sanfey et al., 2003) or deceived by another person (Yin and Weber, 2015). Such negative affect produced by vAI is considered to drive the enforcement of moral norms (Bellucci et al., 2018) such as fairness (Gao et al., 2018) and honesty (Yin et al., 2017; Yin and Weber, 2018). Thus, we hypothesized that the vAI is more engaged in representing the “dirty” personal profits (i.e., expected personal gains from accepting offers in the Bribe vs. Control condition).

On the other hand, we tested how the moral cost of harming a third party, reflected by the expected loss to the third party due to the acceptance of the bribe, was represented in the brain (GLM1b). Substantial evidence has shown that the temporoparietal junction (TPJ) is crucially engaged in representing the mind of others (Schurz et al., 2014), and thus contributes to the trade-off between self- interest and the welfare of others (Hutcherson et al., 2015; Morishima et al., 2012; Obeso et al., 2018). The TPJ is also more active when one’s decision impact a person who is in a disadvantageous condition, such as charity donations (Tusche et al., 2016) and costly helping behavior (Hu et al., 2018). These findings suggested that the TPJ would be sensitive to the moral cost of the bribe-induced financial losses incurred by a third party. We next investigated how these moral costs are integrated with other decision components into a neural value signal during bribery-related decision-making (GLM2). Given the well-established neural account of the ventromedial prefrontal cortex (vmPFC) in value computation (Levy and Glimcher, 2012; Bartra et al., 2013; Ruff and Fehr, 2014), it was natural to predict that vmPFC would be recruited in computing the decision value by integrating various components associated with corruption-related actions (e.g., personal gains and moral costs of taking the bribe).

We also aimed to link the neural signature during bribery-related decision-making to specific choices (i.e., Bribe_(reject – accept)_ – Control_(reject – accept)_, hereafter referred as “the anti-corruption signal”) and to identify brain regions sensitive to the contextual modulation (i.e., Dyad vs. Solo) on the anti-corruption signal (GLM3). The dorsolateral prefrontal cortex (dlPFC) is well known to play a pivotal role in guiding various ethical behaviors concerning fairness (Knoch et al., 2006), justice (Buckholtz et al., 2015), honesty (Zhu et al., 2014; Marechal et al., 2017), and harm (Crockett et al., 2017). Based on these findings, a recent theory posits that the dlPFC is key to flexibly support the pursuit of moral goals in a context-dependent manner (Carlson and Crockett, 2018). Thus, the dlPFC was clearly expected to be engaged in specific choices during bribery-related decision-making and might further be modulated by the specific bribery scenarios.

For all analyses above, a small volume correction (SVC) was conducted within the hypothetical ROIs, as reported in the fMRI data analyses section. Since we did not consider other regions of interest for these analyses, it is unnecessary to correct the number of ROIs in any of our analyses.

As pointed out by the reviewer, we agree that there are other regions that also play some roles in moral behaviors. One of the other “candidate” regions proposed by the reviewer is amygdala, a region typically known as a key hub for processing (negative) basic emotion information (Phelps and LeDoux, 2005). In a recent paper by Garret and others (2016), activity in this region has been found to decrease as dishonest behaviors are repeated (i.e., an adaptation effect of dishonesty). This novel finding was interpreted in terms of an adaptation account, which proposed that the amygdala’s response to an emotion-evoking stimulus weakens with repeated exposure. However, this finding is far from any of the research questions that we aimed to test in our analyses. Moreover, it should be noted that this is the only study so far, to our knowledge, identifying a central role of amygdala in (im)moral behaviors. The same effect was rarely reported in follow-up studies. For these reasons, we did not hypothesize the involvement of amygdala in the current task and we found it was difficult to associate this region with any of our analyses.

Regarding another “candidate” region, the nucleus accumbens (NAcc), this region was typically known as a hub for processing different types of reward (Haber and Knutson, 2009). Consistent with this account, NAcc is found to encode the reward magnitude earned by deception in a recent study (Speer et al., 2020), which was subject to the individual level of cheating behaviors. Although the NAcc was not on the list of our initial interest, we thought that, inspired by this finding, it might also be interesting to examine whether the decision-related neural signal of the NAcc during bribery-related decision-making could predict the corrupt behaviors across individuals (based on GLM1c), in *post-hoc* analyses. To this end, we built up contrasts of bribery-related decision-making for each individual in two separate scenarios (i.e., contrast 1: SB – SC; contrast 2: DB – DC) and the contrast that averages across scenarios (i.e., contrast 3: Bribe – Control), based on GLM1c. Correspondingly, we calculated the acceptance rate in the Bribe condition in each scenario (i.e., SB or DB) or average acceptance rates across scenarios for each individual. Then, we performed a group-level regression analysis with the brain contrast as the dependent measure and the acceptance rate as the predictor, focusing on the NAcc as ROI (defined in the same way as we used for other ROI analyses). However, we did not find that the decision-related NAcc signal specific to corruption was correlated to the acceptance rate of offers in the Bribe condition across individuals in any case (see Author response table 1).

**Author response table 1. resptable1:** Relationship between decision-related NAcc activity specific to corrupt decision-making with acceptance rate of bribes across individuals.

Brain Contrast (Y)	Predictor (X)	Results
SB – SC	Acceptance rate of offers in the SB condition	Positive: No activated voxels Negative: No activated voxels
DB – DC	Acceptance rate of offers in the DB condition	Positive: Peak MNI: 12/2/2 t(37) = 3.28, p(SVC-FWE) = 0.157 Negative: No activated voxels
bribe – control	Mean acceptance rate of offers across SB and DB conditions	Positive: Peak MNI: 6/10/0, t(37) = 2.72, p(SVC-FWE) = 0.243 Negative: No activated voxels

Note: These contrasts were built based on GLM1c. We adopted the voxel-level family-wise error correction (FWE) within the search volume of NAcc (small volume correction, SVC).

Abbreviations: SC: Solo Control, SB: Solo Bribe, DC: Dyad Control, DB: Dyad Bribe.

To sum up, we believe that all predictions concerning specific analyses connecting to specific ROI can be justified based on the evidence mentioned above. With regard to the two “candidate” regions proposed by the reviewer, we argue that their roles in moral decision-making were less typical and actually had little to do with the target behavior and contextual modulators in the current study. Hence we hypothesized neither of them in our analyses.

b) Given the design of the task, any activation contrasting the bribe vs control condition cannot distinguish between representing the proposer lying vs being honest and the computer recommending the low vs high payoff offer, as the proposer always lies when the computer chooses the low payoff. Because of this, I find the results in Figure 3A difficult to interpret. Why would we expect vAI to track expected personal profits positively in the DB condition but negatively in the SC condition, and not at all in the other two conditions? There also seems to be a main effect of scenario (dyad vs solo) on vAI tracking on expected profits, which is not discussed.

There are three key points we would like to clarify. First, we did not perform the contrast between the Bribe and the Control condition using the onset regressor of the decision event. Instead, we compared the parametric modulator (PM) of the expected personal gains between the Bribe and the Control condition. The main focus of GLM1a is to identify, in a power-holder, which brain regions specifically encode the expected personal gains resulting from the acceptance of bribes.

Second, the PM of expected personal gains used in GLM1a did not change depending on the option selected by the computer. Instead, it referred to the amount of money involved in the offer proposed by the proposer, which is always calculated based on a certain proportion of the large payoff reported by the proposer (i.e., the computer chosen option in the Control condition or the computer non-chosen option in the Bribe condition). This ensures values of PM exactly the same in the Bribe and the Control condition. Take the following case in the Solo scenario as example: according to the current study, the fictive proposer (E.L.) could propose an offer based on the reported payoff for the participant (as a power-holder), which was always 96 CNY in either condition as indicated by the blue arrow. (see Figure 1B).

Third, we did not form such a specific prediction about the ventral anterior insula (vAI) as the reviewer mentioned in the comment. What we actually hypothesized is that vAI is more engaged in representing the expected personal gains from accepting offers in the Bribe condition (i.e., PM contrast: Bribe vs. Control), as we delineated in the Introduction based on previous literature. Given this hypothesis, we compared the PM of expected personal gains between the Bribe and the Control condition with a focus on vAI, and found that this region (especially the left side) was indeed more sensitive to the expected personal gains from a fraudulent proposer (i.e., Bribe vs. Control). This result thus conforms to our hypothesis. We extracted the parameter estimates (i.e., contrast values of the activated vAI cluster) only for visualization and we are not trying to interpret why the vAI tracks expected personal profits negatively in the Solo Control (SC) condition.

According to the reviewer’s suggestion, we investigated whether there is a main effect of scenario (Dyad vs. Solo) in vAI. To do this, we compared the PM of expected personal gains between the Dyad and the Solo scenario, within the search volume of vAI. We found that the right vAI displayed an increased sensitivity to the expected personal gains in the Dyad (vs. Solo) scenario (peak MNI: 36/14/-14; t(114) = 3.86, p(SVC-FWE) = 0.022), while the left vAI showed a similar trend that did not reach statistical significance (peak MNI: -36/10/-16; t(114) = 3.09, p(SVC-FWE) = 0.146;). We have now brought up this point in the Discussion, which reads as follows:

“… Interestingly, we also observed that the vAI (especially the right part) is more engaged in encoding expected personal gains in the Dyad (vs. Solo) scenario. The vAI plays a critical role in guiding dishonest decisions under various circumstances (Yin et al., 2017) and in perceiving other’s dishonest intentions (Yin and Weber, 2015). These findings can be broadly linked to the modulation of aversive feelings by vAI, that generate motivation to social norm enforcement (Bellucci et al., 2018). Our results show that a key computation performed by the vAI signal is to encode bribery-related profits, especially when a potential victim is involved in the social context. This signal might reflect an aversive feeling towards moral transgression brought by bribe-taking behavior, which could contribute to (but not necessarily lead to) preventing powerholders from being corrupted.…”

c) Why wasn't the expected loss to the third party added as a parametric modulator of the DB condition in GLM1a, thus allowing to control for the expected gains of the participant and the proposer? Is this because the loss to the third party is highly correlated with the gain of the proposer? If so, then those two signals can't be separated, and this should be addressed.

We chose not to incorporate the PM of expected loss to the third party into the GLM1a because this GLM aims to identify brain regions specifically encoding the expected gains due to bribe-taking (regardless of whether or not a third party is present). For this reason, it is preferable to keep the design matrix balanced so that each onset regressor of the decision event is attached with the exact same PMs (i.e., PM of expected gains for the proposer and the participant). Since the expected loss to the third party only takes place in the Dyad Bribe (DB) condition, adding this PM to GLM1a would leave the design matrix unbalanced between conditions which might bias the results with regard to the main focus of GLM1a.This information has now been added to the Materials and methods.

To comply with the reviewer’s suggestion, we have now investigated the brain regions tracking the expected loss to the third party while controlling for the effect of expected gains (either for the proposer or the participant). To this end, we ran an additional GLM analysis in which we not only added the PM of the expected loss (as we did in GLM1b; the first PM) but also attached the PMs of expected gains for the proposer and the participant to the onset regressor of the decision event in the DB condition (the second and third PM). Here we adopted the default orthogonalization by SPM12 to control for the co-linearity between these PMs. The results showed a significant parametric modulation of the expected loss to the third party in the right TPJ even after controlling for the other two PMs (peak MNI: 54/-48/-4; t(38) = 4.63, p(SVCFWE) = 0.012). We have added this information in the Materials and methods and Results (see Figure 3—figure supplement 2).

d) The rationale for examining vmPFC-dlPFC functional connectivity analyses is unclear to me. Was connectivity with other regions of interest, like TPJ or vAI, tested but not significant? If so, the authors should be clear about this and correct for the number of regions tested. Similarly, the Materials and methods section about the inter-subjects RSA suggests that several "hypothesized regions including bilateral dlPFC" were tested, but then the results focus exclusively on the dlPFC. If other regions were indeed tested, this should be clarified and accounted for.

One of the questions we wanted to address was how is value computation linked to the final decision in a social context involving corrupt acts (i.e., the power-holder’s bribe-taking behavior) at the brain system level. The previous literature strongly suggested a crucial role of the vmPFC in value computation (Levy and Glimcher, 2012; Bartra et al., 2013; Ruff and Fehr, 2014) and of the dlPFC in guiding various types of decisions (Tanji and Hoshi, 2008; Figner et al., 2010; Ruff et al., 2013; Marechal et al., 2017). Importantly, recent work has highlighted the modulatory role of dlPFC on the value signal in vmPFC, especially when the decision-making process requires individuals to employ self-control to inhibit the impulse to choose immediate rewards (vs. long-term rewards; Hare et al., 2009; 2014) or personal profits (vs. moral values; Baumgartner et al., 2011; Dogan et al., 2016). Based on this evidence, we hypothesized that we might observe a stronger vmPFC-dlPFC functional coupling during the decision period in the Bribe (vs. control) condition, in which participants as power-holders need more self-control to overcome the lure of accepting bribes that result in moral costs. Hence, we tested this hypothesis using psycho-physiological interaction (PPI) with vmPFC as the seed region and dlPFC as the only region of interest (ROI). Since we did not hypothesize that the functional coupling between vmPFC and other regions (e.g., TPJ or vAI) changed during bribery-related decision-making, we did not implement the analyses using these regions as ROIs.

Similarly, in the inter-subject representational similarity analysis (IS-RSA), we explicitly treated it as an exploratory analysis because IS-RSA is a fairly novel methodological approach in the neuroscience literature. Nevertheless, we still focused on the dlPFC in this analysis because recent evidence on social and moral decisionmaking indicated that the decision-related signal in dlPFC could be modulated by individual variations of self-serving dishonesty (Dogan et al., 2016; Yin and Weber, 2018) and harm aversion (Crockett et al., 2017), which are related with the two types of moral costs measured in the current task.

We have incorporated more detailed statements about the rationale underlying our hypotheses and analyses in the Introduction, which reads as follows:

“… More intriguingly, recent work has highlighted the modulatory role of dlPFC on the value signal in vmPFC when the decision-making process requires individuals to exert self-control to inhibit the impulse to choose immediate rewards (vs. long-term rewards; Hare et al., 2009; Hare et al., 2014) or personal profits (vs. moral values; Baumgartner et al., 2011; Dogan et al., 2016). […] Hence, we performed an exploratory analysis, again with a focus on the dlPFC given the evidence above, to probe whether such a relationship exists, in a bribery setting, by applying a multivariate approach.”

e) The interpretation of the IS-RSA results in the Discussion is unclear, especially in what those results mean in and of themselves, and how they can be reconciled with the univariate dlPFC result and the vmPFC-dlPFC functional connectivity analysis.

We thank the reviewer for pointing this out, and we agree that it is necessary to add more details with regard to the methodological rationale behind IS-RSA which helps to improve the understanding of this result. As a novel analytical approach, IS-RSA applies RSA to uncover the neural-behavioral relationship across individuals. Compared with the mass-univariate approach, IS-RSA allows us to associate multidimensional behavioral measures with a geometric representation of information based on multi-voxel neural patterns across individuals, rather than simply linking a single behavioral measure with averaged activities across voxels in a certain region (Kriegeskorte et al., 2008; Popal et al., 2020).

Here, with the help of IS-RSA, we were able to map differences in neural signals during bribery-related decision-making (i.e., Bribe vs. Control) directly onto our behavioral model characterizing corrupt behaviors of a power-holder, which contains two key parameters capturing different types of moral costs (i.e., θ: the moral cost brought by conniving with a fraud committed by a proposer; ω: the moral cost brought by harming the interest of a third party). Given the previous literature (see our reply to comment 2d), we focused on dlPFC and tested whether the neural patterns during bribery-related decision-making in dlPFC were similar for participants who showed similar context-dependent corrupt behaviors, reflected by bribery-specific preferences (i.e., an RDM built upon θ and ω across individuals). Our results showed that this is indeed the case, which provides novel evidence for the role of dlPFC in representing geometric information concerning a multidimensional model of moral preferences across participants.

In our view, all these dlPFC-related results provide evidence for a critical role of dlPFC in multiple aspects of bribery-related decision-making. The univariate result (based on GLM3), from a within-subject perspective, showed an increased anticorruption dlPFC signal in the Dyad (vs. Solo) scenario. This result suggests that dlPFC plays a specific role in guiding the choice of rejecting bribes that specifically involved harm to a third party. Both the functional connectivity (PPI) results and the IS-RSA results concerned the between-subject perspective. Compared with the IS-RSA result (see our interpretations above), the PPI result provides evidence from a brain network angle and suggests that the modulatory role of dlPFC on the value signal in vmPFC during bribery-related decision-making was sensitive to individual variations when considering the moral cost brought by conniving with a fraud committed by a proposer. In general, these findings indicate that the dlPFC signature, or pattern, might reflect the preference to take or refuse a bribe across individuals as power-holders.

Following the reviewer’s suggestions, we have incorporated these points in the revised manuscript accordingly (Discussion), which hopefully clarify the interpretation of the IS-RSA results and its relationship with the PPI results. Now it reads as follows:

“… Notably, all these results provide evidence for a critical role of the dlPFC in different aspects of bribery-related decision-making. The univariate result (based on GLM3) suggests that, from a within-subject perspective, the dlPFC plays a dedicated role in guiding the choice to reject bribes in a context-dependent manner. Both the functional connectivity results and the IS-RSA results concern the between-subject perspective. These findings indicate that the dlPFC signature (or pattern) might reflect the preference of whether to accept or refuse a bribe across individuals as powerholders. These results are consistent with a causal role of the dlPFC in ethical behaviors (Maréchal et al., 2017; Zhu et al., 2014), and may inspire future studies to investigate whether such a causal role extends to corrupt behaviors and how it varies between individuals.”

Reviewer #2:The reported study investigates the neural basis of corruption in social interactions. The main finding is that two types of moral costs in bribery (norm violation versus harming a third-party) correlate with activation in dissociable brain regions (insula versus TPJ). There is much to like about this well-written manuscript: it addresses an important question, and the experimental manipulations appear sound. The manuscript will certainly be of interest to a broad readership working on social interactions or on the neural basis of decision making. Nevertheless, I have a couple of concerns, particularly regarding data analysis, which the authors need to address in a revision.1) The GLMs used for the fMRI analyses should be specified in more detail already in the Results section, not only in the Materials and methods. This would make it easier to understand which contrasts show significant activation in insula or TPJ. In general, I find the presentation of the imaging results rather confusing, mainly because a large number of analyses was computed (nine in total: GLM1a-c, GLM2a-b, GLM3, PPI, multivariate analysis, IS-RSA), and the authors often just selectively report one contrast of each GLM, while the results of other contrasts are not even reported in supplementary tables. This analysis approach raises questions regarding the robustness of the imaging results. For example, when testing the hypothesis that the TPJ shows enhanced activation in dyad versus solo scenarios, no significant results are observed in two contrasts for GLM1c, but only in an additional multivariate analysis. However, these non-significant results are ignored in the Discussion section, and the significant TPJ finding in the multivariate analysis is taken as evidence for the authors' hypothesis. It also remains unclear why the authors specified separate models with parametric modulators for personal benefits (GLM1a) and third-party loss (GLM1b), as in principle these variables could be modelled within one model. All this leaves the impression of cherry-picking and makes the imaging results appear less robust and convincing than as they are presented.

We thank the reviewer for this suggestion. Following this suggestion, we have updated the relevant part before we introduced each of our fMRI results to specify the details of GLMs used for the fMRI analyses, which reads as follows:

“We implemented the general linear model (GLM) analyses to test specific hypotheses concerning different research questions (see Materials and methods for details of GLM analyses). […] To simplify the analysis, we computed the neural activity specific to rejecting as opposed to accepting offers (i.e., reject vs. accept) in all four conditions and then defined the anti-corruption neural signals with such rejection-specific neural activity in the Bribe condition (i.e., contrast: Bribe_(reject – accept)_ – Control_(reject – accept)_)…”.

With regard to the rationale underlying our analyses mentioned in the current manuscript, we argued that each analysis has its own goal with a clear hypothesis, based on previous literature. Briefly, both GLM1a and GLM1b investigated how different forms of moral costs were encoded in the brain of a power-holder. More specifically, GLM1a tested whether the moral cost of conniving with a fraudulent proposer altered the valuation of the expected personal gains, due to the acceptance of the bribe. This was focused on the vAI as the region of interest (ROI). GLM 1b examined how the moral cost of harming a third party, reflected by the expected loss to the third party due to the acceptance of the bribe, was represented in the brain, with a focus on the TPJ as ROI. The decoding analysis was only used to corroborate the TPJ finding from a multivariate perspective. GLM2 aimed to investigate how these moral costs were integrated with other decision components into a neural value signal during bribery related decision-making, with a focus on vmPFC. GLM3 also aimed to link the neural signature during bribery-related decision-making to specific choices (i.e., accept or reject) and examine how this neural signal was modulated by the scenario (i.e., Dyad vs. Solo), with a focus on the dlPFC. While the GLM analyses focused more on the within-subject experimental effect on local activations, the PPI analysis was planned to examine the functional coupling between brain regions (i.e., vmPFC and dlPFC) during bribery-related decision-making (Bribe vs. Control) from a network perspective. Lastly, the IS-RSA concerned the between-subject effect, which allowed us to map differences in neural signals of bribery-related decision-making (Bribe vs. Control) directly onto a multi-dimensional space of model parameters that characterized the context-dependent corrupt behaviors. Although this latter analysis is exploratory, we still focused on the dlPFC, given its critical role in representing moral preferences and behaviors across individuals, as proposed in the literature, taking a univariate approach.

While it is theoretically possible to model all these parametric modulators (PM) within one GLM, we chose not to do so because GLM1a aims to identify brain regions specifically encoding the expected gains due to bribe-taking. For this reason, it would be better to keep the design matrix balanced so that each onset regressor of the decision event is attached with the exact same parameter modulators (i.e., PM of expected gains for the proposer and the participant). Since the expected loss to the third party only took place in the Dyad Bribe (DB) condition, adding this PM to GLM1a would leave the design matrix unbalanced between conditions which might bias the results. However, as the reviewer suggested, we performed an additional GLM analysis and found that the right TPJ tracked the expected loss to the third party, after controlling for the effect of expected gains for either the proposer or the participant (see Figure 3—figure supplement 2 and Supplementary File 1F).

Regarding the missing GLM contrasts, it is highly likely that no region survived even under a lenient whole-brain threshold (*p* < 0.001 uncorrected at the voxel-level) so that we had no significant results to report. However, we agree that this might cause the impression of cherry-picking, and thus we have reported all possible GLM contrasts in Supplementary File 1F-1I.

2) Regarding the mixed linear models for the analysis of behavioral data, the authors state that "factors allowing varying intercept across participants" were entered as random-effect predictors. Please be more precise regarding which random effects were specified in the model. It is generally recommended to maximize the random effects structure in order to minimize the risk of type I errors (Barr et al., 2013, Journal of Memory and Language). As the current study seems to follow a strict within-subject design, I think that all fixed-effect predictors should be modelled as random slopes in addition to random intercepts

We agree with the reviewer’s suggestion and have re-done all the mixed-effect analyses as suggested (see the Materials and methods and the Results).

3) The task did not involve real social interactions, but the offers were computer-generated. I might have missed it, but it seems nowhere stated whether participants believed the cover story or not (was this assessed at all). In any case, the authors should add a caveat in the Discussion section clarifying that the social interactions in the study were only hypothetical.

As the reviewer pointed out, the present study did not adopt a task involving real interactions but a multi-round, single-shot economic game instead. This allowed us to treat each decision independently and thus to rule out the confounding effect of learning and reputation concerns via repeated interactions with the same partner. Since this concern was also raised by reviewer #1, please find our responses above that explains why we did not formally test whether participants believed our cover story and what evidence that converges to indicate that the participants were likely to have believed the experimental setting.

Despite that, we agree with the reviewer that the lack of real social interaction in the present study is an important point and we have added it as a caveat to the Discussion.

Reviewer #3:Dr. Hu et al. report a neuroimaging study of corruption in which (computer agent) proposers provide participants the opportunity to personally benefit from turning a blind eye to deception that in some cases has monetary costs for a third party. Through a series of computational models, the authors confirm that participants incur a moral cost, beyond inequity models, for engaging in the corrupt act, that is at its worst when a third party is injured. Each component of the model is then tied to a specific aspect of brain function that aligns with prior findings. The authors conclude that an inhibition mechanism is associated with reduced participation in corrupt acts.I enjoyed the paper. It covers an interesting topic, is methodical, and shows a great deal of expertise in a wide range of methods.1) Tests for the involvement of a particular brain region in a given step of the corruption decision are mixed without justification. GLM and MVPA analyses seem to be deployed as tests with increasing sensitivity rather than to test for computational differences. Not making a distinction is understandable since there is not a consensus on how each should be interpreted. However, this sort of variation on testing until a finding is reached can lead to overestimation of effect sizes and confirmation bias of previously proposed roles. Is there a reason beyond sensitivity for using multivariate models in some cases rather than others?

We apologize for the lack of justification of the multivariate analyses used in the present study. In fact, we adopted this approach to address two specific questions. The first, (i.e., the decoding analysis), concerns whether the temporo-parietal junction (TPJ) extending to the posterior temporal sulcus (pSTS) was selectively engaged in briberyrelated decision-making that involved harm (i.e., financial losses) incurred by a third party (i.e., the Dyad scenario). Notably, this analysis was originally implemented as a supplementary analysis to corroborate the univariate finding (see Figure 3B). The second, (i.e., the inter-subject representational similarity analysis, IS-RSA), concerns how the preferences for bribery, characterized by two parameters, reflecting different forms of moral costs brought by bribe-taking behaviors derived from a computational model, were represented in the brain across individuals.

It is indeed correct that multivariate analyses, compared with the univariate analyses, generally use more information and thus increase the sensitivity to detect meaningful differences (Hebart and Baker, 2018; Haynes, 2015). However, this is not the main reason that led us to adopt multivariate analyses. In fact, these different approaches were used because they provide rich results that complement each other. In particular, the decoding analysis revealed that TPJ/pSTS work differently during bribery-related decision-making between the Solo and the Dyad scenario by showing distinct activation patterns (but possibly not the mean activation intensity across voxels).

Moreover, the IS-RSA allowed us to map differences in neural signals of bribery-related decision-making (i.e. with a focus on the dlPFC) directly onto a multi-dimensional space of model parameters that characterize the context-dependent corrupt behaviors. Taken together, we believe the use of multivariate approaches, together with other analyses, provides novel evidence regarding the role of TPJ/pSTS and dlPFC in complex social behaviors.

2) A number of GLMs are used to reach different conclusions presumably because of covariance issues between regressors. This, in and of itself, is not a problem if the covariance tables are shown and the repartitioning of common variance is acknowledged/interpreted.

It is perfectly true that due to the present design, some of the payoff-related parametric modulators (PM) are correlated (see Supplementary File 1K). Although multi-colinearity between predictors (PMs) biases the estimation of the GLM to some extent and was indeed one reason driving us to implement separate GLMs, the other reason for not incorporating all payoff-related PMs into one GLM1 was that GLM1a aims to identify brain regions that specifically encode the expected gains due to bribetaking behavior. Therefore, we decided to keep the design matrix balanced so that each onset regressor of the decision event was attached with the exact same PMs (i.e., the PM of expected gains for the proposer and the participant). Since the expected loss to the third party only took place in the Dyad Bribe (DB) condition, adding this PM to GLM1a would leave the design matrix unbalanced between conditions which might bias the results (see the Materials and methods and the Results).

As suggested by the reviewer, we have incorporated the correlation table between three key payoff-related PMs in Supplementary File 1K and acknowledged this point in the Materials and methods accordingly.

3) Individual difference model – individual difference studies of forty participants assume a large effect size to be considered reliable. Most psychological (and biological indices) do not fall in this range. The result is fine as an exploratory analysis but that means it should be described as such.

We reported two analyses with regard to individual differences. Although we agree that the results of inter-individual difference analyses may not be reliable given the sample size, we implemented these analyses with specific goals. As mentioned earlier, we explicitly framed the IS-RSA as an exploratory analysis, because it is a novel analytical approach that emerges in recent fMRI literature. However, we still focused on the dlPFC in this analysis because recent evidence reveals the link between decisionrelated signals in dlPFC and individual variations of self-serving dishonesty (Dogan *et al.*, 2016; Yin and Weber, 2018) and harm aversion (Crockett *et al.*, 2017), which are related with the two types of moral costs measured in the current task (see the Introduction).

Unlike the IS-RSA, we actually had a clear prediction in the functional connectivity analysis using PPI, that the vmPFC-dlPFC connectivity would be enhanced during bribery-related decision-making (Bribe vs. Control). This prediction was based on previous literature concerning economic and social decision-making that requires selfcontrol (Hare et al., 2009; 2014; Baumgartner *et al.*, 2011; Dogan *et al.*, 2016). Although our expected result was not observed, we showed that such vmPFC-dlPFC connectivity during bribery-related decision-making was modulated by individual differences in the moral cost of conniving with the fraudulent briber. Taken together, we have clarified this point and revised statements in the Introduction and Materials and methods accordingly.

4) Deception success measures. Was any data collected to confirm the believability of the proposer and third-party deception? Were participants debriefed afterwards?

We did not formally test whether participants believed our cover story in the present study. Since this concern was also raised by the other two reviewers, please find our detailed responses above, explaining why we did not do so and what evidence converges to indicate that they were very likely to have believed the cover story, to avoid unnecessary repetition. Despite that, we agree that this is an important point to address and we have thus added it to the Discussion accordingly.

As mentioned in the Materials and methods, we did debrief the participants at the very end of the experiment.

5) The brain data is largely used to lend construct validity to the corruption task and confirm psychological interpretations of the processes involved in the decision. Ideally, the neural data would be used to adjudicate between two competing behavioral models (and/or behavioral data used to adjudicate between competing neural models). This somewhat lessens the utility of a promising neuroimaging dataset.

As the reviewer pointed out, the neural imaging data were mainly used to identify the brain regions (signals or patterns) involved in different aspects during briberyrelated decision-making. Critically, these data were linked in two ways to the behavioral model of context-dependent corrupt acts. First, we observed a strong vmPFC value signal, given the winning model, during decision-making in both the Bribe and the Control condition. This result can be considered as neural evidence that validates our behavioral model. Second, we identified a potential role of dlPFC in representing bribery-related preferences, characterized by two model-based parameters reflecting different forms of moral costs brought by bribe-taking behaviors, across individuals. Although it would be ideal to adjudicate the behavioral model with neural data or *vice versa*, we found it difficult to apply this set of analyses to the present case because our behavioral models were mainly based on economic utility models which all assumed the construction of a (subjective) decision value in a certain way, and were less informed by the neural data.

6) Complete analysis scripts (Main and SI, behavioral and neuroimaging) and minimally processed imaging data should be posted and referenced for the article. It is not clear that these fit in the 'source data' option for eLife without an accession number noted in the article.

We fully embrace the open-and-transparency policy advocated by *eLife* and have already uploaded the most relevant codes and data to an open data depository (https://github.com/huyangSISU/2021_Corruption) which will be available to the public once the paper is accepted.